# SAMPa: Sharpness-aware Minimization Parallelized

**Wanyun Xie**
EPFL (LIONS)
`wanyun.xie@epfl.ch`

**Thomas Pethick**
EPFL (LIONS)
`thomas.pethick@epfl.ch`

**Volkan Cevher**
EPFL (LIONS)
`volkan.cevher@epfl.ch`

## Abstract

Sharpness-aware minimization (SAM) has been shown to improve the generalization of neural networks. However, each SAM update requires *sequentially* computing two gradients, effectively doubling the per-iteration cost compared to base optimizers like SGD. We propose a simple modification of SAM, termed SAMPa, which allows us to fully parallelize the two gradient computations. SAMPa achieves a twofold speedup of SAM under the assumption that communication costs between devices are negligible. Empirical results show that SAMPa ranks among the most efficient variants of SAM in terms of computational time. Additionally, our method consistently outperforms SAM across both vision and language tasks. Notably, SAMPa theoretically maintains convergence guarantees even for *fixed* perturbation sizes, which is established through a novel Lyapunov function. We in fact arrive at SAMPa by treating this convergence guarantee as a hard requirement—an approach we believe is promising for developing SAM-based methods in general. Our code is available at `https://github.com/LIONS-EPFL/SAMPa`.

## 1 Introduction

The rise in deep neural network (DNN) usage has spurred a resource examination of training optimization methods, particularly focusing on bolstering their *generalization ability*. Generalization refers to a DNN's proficiency in effectively processing and responding to new, previously unseen data originating from the same distribution as the training dataset. A DNN with robust generalizability can reliably perform well on real-world tasks, when confronted with novel data instances or when quantized.

Improving generalization poses a significant challenge in machine learning. Recent studies suggest that smoother loss landscapes lead to better generalization [Keskar et al., 2017, Jiang* et al., 2020]. Motivated by this concept, *Sharpness-Aware Minimization (SAM)* has emerged as a promising optimization approach [Foret et al., 2021, Zheng et al., 2021, Wu et al., 2020b]. It is the current state-of-the-art to seek flat minima by solving a min-max optimization problem, in which the inner maximizer quantifies the sharpness as the maximized change of training loss and the minimizer both the vanilla training loss and the sharpness. As a result, SAM significantly improves the generalization ability of the trained DNNs which has been observed across various supervised learning tasks in both vision and language domains [Foret et al., 2021, Bahri et al., 2021, Zhong et al., 2022]. Moreover, some variants of SAM improve its generalization further [Kwon et al., 2021, Kim et al., 2022].

Although SAM and some variants achieve remarkable generalization improvement, they increase the computational overhead of the given base optimizers. In SAM algorithm [Foret et al., 2021], each update consists of two forward-backward computations: one for computing the perturbation and the other for computing the update direction. Since these two computations are not parallelizable, SAM doubles the training time compared to the standard empirical risk minimization (ERM).

Several variants of SAM have been proposed to improve its efficiency. A common strategy involves integrating SAM with base optimizers in an alternating fashion like RST [Zhao et al., 2022b], LookSAM [Liu et al., 2022], and AE-SAM [Jiang et al., 2023]. Moreover, ESAM [Du et al., 2022a]

uses fewer samples and updates fewer parameters to decrease the computational cost. However, some of these algorithms are suboptimal and their computational time overhead cannot be ignored completely. Du et al. [2022b] utilize loss trajectory instead of a single ascent step to estimate sharpness, albeit at the expense of memory consumption due to the storage of historical outputs or past models.

Since the runtime of SAM critically depends on the sequential computation of its gradients, we ask

*Can we perform these two gradient computations in parallel?*

In the sequel, we will answer this question in the affirmative. Note that since the second gradient computation highly depends on the first one seeking the worst case around the neighborhood, it is challenging to break the sequential relationship between two gradients in one update.

To this end, we introduce a new optimization sequence that allows us to parallelize these two gradient computations completely. Furthermore, we also integrate the optimistic gradient descent method with our parallelized version of SAM. Our final algorithm, named SAMPa, not only allows for a theoretical speedup up to $2\times$ when there is no communication overhead but also improves the generalization further. Specifically, we make the following contributions:

- **Parallelized formulation of SAM.** We propose a novel parallelized solution for SAM, which breaks the sequential nature of two gradient computations in each SAM's update. It enables the simultaneous calculation of both gradients, potentially halving the computational time compared to vanilla SAM. We also integrate this parallelized method with the optimistic gradient descent method, known for its stabilizing properties, finalized to SAMPa.

- **Convergence guarantees.** Our theoretical analysis establishes a novel Lyapunov function, through which we prove convergence guarantees of SAMPa even with a *fixed* perturbation size. We arrive at SAMPa by treating this convergence guarantee as a hard requirement, which we believe is promising for developing other SAM-based methods.

- **Improved generalization and efficiency.** Our numerical evidence shows that SAMPa significantly reduces overall computational time even with a basic implementation while achieving superior generalization performance. Indeed, SAMPa requires the least computational time compared to the other four efficient SAM variants while enhancing generalization across different tasks. Notably, the relative improvement from SAM to SAMPa is 62.07% on CIFAR-10 and 32.65% on CIFAR-100, comparable to the gains from SGD to SAM. SAMPa also shows benefits on a large-scale dataset (ImageNet-1K), image and NLP fine-tuning tasks, as well as noisy label tasks, with the capability to integrate with other SAM variants.

## 2  Background and Challenge of SAM

This section starts with a brief introduction to SAM and its sequential nature of gradient computations. Subsequently, we discuss naive attempts including an approach from existing literature and our initial attempt which serve as essential motivation for constructing our final algorithm in the next section.

### 2.1  SAM and its challenge

Motivated by the concept of minimizing sharpness to enhance generalization, SAM attempts to enforce small loss around the neighborhood in the parameter space [Foret et al., 2021]. It is formalized by a minimax problem

$$\min_x \max_{\epsilon:\|\epsilon\|\leq\rho} f(x+\epsilon) \tag{1}$$

where $f$ is a model parametrized by a weight vector $x$, and $\rho$ is the radius of considered neighborhood.

The inner maximization of Equation (1) seeks for maxima around the neighborhood. To address the inner maximization problem, Foret et al. [2021] employ a first-order Taylor expansion of $f(x+\epsilon)$ with respect to $\epsilon$ in proximity to 0. This approximation yields:

$$\epsilon^\star = \arg\max_{\epsilon:\|\epsilon\|\leq\rho} f(x+\epsilon) \approx \arg\max_{\epsilon:\|\epsilon\|\leq\rho} f(x) + \langle \nabla f(x), \epsilon \rangle = \rho \frac{\nabla f(x)}{\|\nabla f(x)\|} \tag{2}$$

SAM first obtains the perturbed weight $\widetilde{x} = x + \epsilon^\star$ by this approximated worst-case perturbation and then adopts the gradient of $\widetilde{x}$ to update the original weight $x$. Consequently, the updating rule at each iteration $t$ during practical training is delineated as follows:

$$\widetilde{x}_t = x_t + \rho \frac{\nabla f(x_t)}{\|\nabla f(x_t)\|}, \quad x_{t+1} = x_t - \eta_t \nabla f(\widetilde{x}_t) \tag{SAM}$$

It is apparent from SAM, that the update requires two gradient computations for each iteration, which are on the clean weight $x_t$ and the perturbed weight $\widetilde{x}_t$ respectively. These two computations are *not parallelizable* because the gradient at the perturbed point $\nabla f(\widetilde{x}_t)$ highly depends on the gradient $\nabla f(x_t)$ through the computation of the perturbation $\widetilde{x}_t$. Therefore, SAM doubles the computational overhead as well as the training time compared to base optimizers *e.g.,* SGD.

## 2.2 Naive attempts

The computational overhead of SAM is primarily due to the first gradient for computing the perturbation as discussed in Section 2.1. *Can we avoid this additional gradient computation?* Random perturbation offers an alternative to the worst-case perturbation in SAM, as made precise below:

$$\widetilde{x}_t = x_t + \rho \frac{e_t}{\|e_t\|} \quad \text{with} \quad e_t \sim \mathcal{N}(0, I)$$
$$x^{t+1} = x^t - \eta_t \nabla f(\widetilde{x}_t) \tag{RandSAM}$$

Unfortunately, it has been demonstrated empirically that RandSAM does not perform as well as SAM [Foret et al., 2021, Andriushchenko and Flammarion, 2022]. The poor performance of RandSAM is maybe not surprising, considering that RandSAM does not converge even for simple convex quadratics as demonstrated in Figure 1.

We argue that the algorithm we construct should at least be able to solve the original minimization problem. Recently, Si and Yun [2024, Thm. 3.3] very interestingly proved that SAM converges for convex and smooth objectives even with a *fixed* perturbation size $\rho$. Fixed $\rho$ is interesting to study, firstly, because it is commonly used and successful in practice [Foret et al., 2021, Kwon et al., 2021]. Secondly, convergence results relying on decreasing perturbation size are usually agnostic to the direction of the perturbation [Nam et al., 2023, Khanh et al., 2024], so the results cannot distinguish between RandSAM and SAM, which behaves strikingly different in practice.

The fact that SAM uses the gradient direction $\nabla f(x_t)$ in the perturbation update, turns out to play an important role when showing convergence. It is thus natural to ask whether another gradient could be used instead. Inspired by the reuse of past gradients in the optimistic gradient method [Popov, 1980, Rakhlin and Sridharan, 2013, Daskalakis et al., 2017], an intuitive attempt is using the previous gradient at the perturbed model, such that $\nabla f(y_t) = \nabla f(\widetilde{x}_{t-1})$, as outlined in the following update:

$$\widetilde{x}_t = x_t + \rho \frac{\nabla f(\widetilde{x}_{t-1})}{\|\nabla f(\widetilde{x}_{t-1})\|}, \quad x_{t+1} = x_t - \eta_t \nabla f(\widetilde{x}_t) \tag{OptSAM}$$

Notice that only one gradient computation is needed for each update. However, the empirical findings detailed in Appendix B.1 reveal that OptSAM fails to match SAM and even performs worse than SGD. In fact, such failure is already apparent in a simple toy example demonstrated in Figure 1, where OptSAM fails to converge. It is not surprising to see its failure. To be specific, in contrast with the optimistic gradient method, $\widetilde{x}_t$ in OptSAM represents an ascent step from $x_t$ while $x_{t+1}$ denotes a descent step from $x_t$, making $\nabla f(\widetilde{x}_t)$ a poor estimate of $\nabla f(x_{t+1})$. In the subsequent Section 3 we detail a principled way of correcting this issue by developing SAMPa.

**Toy example.** We use a toy example $f(x) = \|x\|^2$ to test if an algorithm can be optimized. We show the convergent performance of SAM, two naive attempts in this section, and SAMPa-$\lambda$ that is our algorithm proposed in Section 3. The results in Figure 1 demonstrate that RandSAM and OptSAM fail to converge, whereas SAM and SAMPa-$\lambda$ converge successfully.

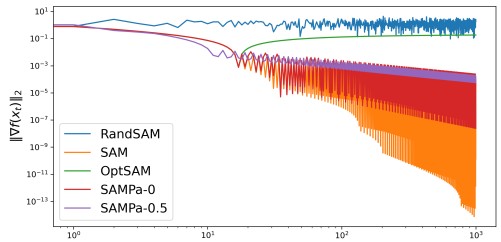

Figure 1: Comparison on $f(x) = \|x\|^2$.

---

**Algorithm 1** SAM Parallelized (SAMPa)

---

**Input:** Initialization $x_0 \in \mathbb{R}^d$, initialization $y_0 = x_0$ and $g_0 = \nabla f(y_0, \mathcal{B}_0)$, iterations $T$, step sizes
$\qquad \{\eta_t\}_{t=0}^{T-1}$, neighborhood size $\rho > 0$, interpolation ratio $\lambda$.

**1 for** $t = 0$ **to** $T - 1$ **do**

**2** $\quad$ Keep minibatch $\mathcal{B}_t$ and sample minibatch $\mathcal{B}_{t+1}$.

**3** $\quad$ Compute perturbed weight $\widetilde{x}_t = x_t + \rho \frac{g_t}{\|g_t\|}$.

**4** $\quad$ Compute the auxiliary sequence $y_{t+1} = x_t - \eta_t g_t$.

**5** $\quad$ Compute gradients $\widetilde{g}_t = \nabla f(\widetilde{x}_t, \mathcal{B}_t)$ and $g_{t+1} = \nabla f(y_{t+1}, \mathcal{B}_{t+1})$ in parallel.

**6** $\quad$ Obtain the final gradient $G_t = (1 - \lambda)\widetilde{g}_t + \lambda g_{t+1}$.

**7** $\quad$ Update weights $x_{t+1} = x_t - \eta_t G_t$.

---

## 3 SAM Parallelized (SAMPa)

As discussed in Section 2.2, we wish to ensure that our developed (parallelizable) SAM variant maintains convergence in convex smooth problems even when using a *fixed* perturbation size. To break the sequential nature of SAM, we seek to replace the gradient $\nabla f(x_t)$ with another gradient $\nabla f(y_t)$ computed at some auxiliary sequence $(y_t)_{t \in \mathbb{N}}$. Provided the importance of the gradient direction $\nabla f(x_t)$ in the convergence proof of SAM, we are interested in picking the sequence $(y_t)_{t \in \mathbb{N}}$ such that the difference $\|\nabla f(x_t) - \nabla f(y_t)\|$ can be controlled. Additionally, we need to ensure that $\nabla f(\widetilde{x}_t)$ and $\nabla f(y_{t+1})$ can be computed in parallel.

Considering these design constraints, we arrive at the SAMPa method that is similar to SAM apart from the gradient used in perturbation calculation is computed at the auxiliary sequence $(y_t)_{t \in \mathbb{N}}$, as illustrated in the following update:

$$
\begin{aligned}
\widetilde{x}_t &= x_t + \rho \frac{\nabla f(y_t)}{\|\nabla f(y_t)\|} \\
y_{t+1} &= x_t - \eta_t \nabla f(y_t) \\
x_{t+1} &= x_t - \eta_t \nabla f(\widetilde{x}_t)
\end{aligned}
\tag{SAMPa}
$$

where the particular choice $y_{t+1}$ is a direct consequence of the analysis, as discussed in Appendix C. Importantly, $\nabla f(\widetilde{x}_t)$ and $\nabla f(y_{t+1})$ can be computed in parallel in this case. Intuitively, if $\nabla f(y_t)$ and $\nabla f(x_t)$ are not too different then the scheme will behave like SAM. This intuition will be made precise by our potential function used in the analysis of Section 4.

In SAMPa the gradient at the auxiliary sequence $\nabla f(y_{t+1})$ is only used for the perturbation update. It is reasonable to ask whether the gradient can be reused elsewhere in the update. As $y_{t+1}$ can be viewed as an extrapolated sequence of $x_t$, it is directly related to the optimistic gradient descent method [Popov, 1980, Rakhlin and Sridharan, 2013, Daskalakis et al., 2017] as outlined below:

$$
y_{t+1} = x_t - \eta_t \nabla f(y_t), \quad x_{t+1} = x_t - \eta_t \nabla f(y_{t+1})
\tag{OptGD}
$$

This celebrated scheme is known for its stabilizing properties as made precise through its ability to converge even for minimax problems. By simply taking a convex combination of these two convergent schemes, $x_{t+1} = (1-\lambda)\,\mathrm{SAMPa}(x_t) + \lambda\,\mathrm{OptGD}(x_t)$, we arrive at the following update rule:

$$
\boxed{
\begin{aligned}
\widetilde{x}_t &= x_t + \rho \frac{\nabla f(y_t)}{\|\nabla f(y_t)\|} \\
y_{t+1} &= x_t - \eta_t \nabla f(y_t) \\
x_{t+1} &= x_t - \eta_t (1 - \lambda) \nabla f(\widetilde{x}_t) - \eta_t \lambda \nabla f(y_{t+1})
\end{aligned}
}
\tag{SAMPa-$\lambda$}
$$

where $\lambda \in [0, 1]$. Notice that SAMPa is obtained as the special case SAMPa-0 whereas SAMPa-1 recovers OptSAM. Importantly, SAMPa-$\lambda$ still admits parallel gradient computations and requires the same number of gradient computations as SAMPa.

**SAMPa with stochasticity.** An interesting observation in the SAM implementation is that both gradients for perturbation and correction steps have to be computed on the same batch; otherwise, SAM's performance may deteriorate compared to the base optimizer. This is validated by our

empirical observation in Appendix B.2 and supported by [Li and Giannakis, 2024, Li et al., 2024]. Therefore, we need to be careful when deploying SAMPa in practice.

Considering the stochastic setting, we present the finalized algorithm named SAMPa in Algorithm 1. Note that $\widetilde{g}_t = \nabla f(\widetilde{x}_t, \mathcal{B}_t)$ represents the stochastic gradient estimate of the model $\widetilde{x}_t$ on mini-batch $\mathcal{B}_t$, and similarly $g_{t+1} = \nabla f(y_{t+1}, \mathcal{B}_{t+1})$ is the gradient of the model $y_{t+1}$ on mini-batch $\mathcal{B}_{t+1}$. This ensures that the gradient $g_t$, used to calculate the perturbed weight $\widetilde{x}_t$ (line 3), is computed on the same batch as the gradient $\widetilde{g}_t$. As demonstrated in line 5, SAMPa also requires 2 gradient computations for each update. Despite this, SAMPa only needs half of the computational time of SAM because $\widetilde{g}_t$ and $g_{t+1}$ are calculated in parallel.

## 4 Analysis

In this section, we will show convergence of SAMPa even with a *nondecreasing* perturbation radius. The analysis relies on the following standard assumptions.

**Assumption 4.1.** *The function $f : \mathbb{R}^d \to \mathbb{R}$ is convex.*

**Assumption 4.2.** *The operator $\nabla f : \mathbb{R}^d \to \mathbb{R}^d$ is L-Lipschitz with $L \in (0, \infty)$, i.e.,*
$$\|\nabla f(x) - \nabla f(y)\| \le L\|x - y\| \quad \forall x, y \in \mathbb{R}^n.$$

The direction of the gradient used in the perturbation turns out to play a crucial role in the analysis. Specifically, we will show that the auxiliary gradient $\nabla f(y_t)$ in SAMPa can safely be used as a replacement of the original gradient $\nabla f(x_t)$ in SAM, since we will be able to control their difference. This is made precise by the following potential function used in our analysis:
$$\mathcal{V}_t := f(x_t) + \tfrac{1}{2}(1 - \eta_t L)\|\nabla f(x_t) - \nabla f(y_t)\|^2.$$

As the potential function suggests we will be able to telescope the last term, which means that our convergence will remarkably only depend on the *initial* difference $\|\nabla f(y_0) - \nabla f(x_0)\|$, whose dependency we can remove entirely by choosing the initialization as $x_0 = y_0$. See the proof of Theorem 4.4 for details. In the following lemma we establish descent of the potential function.

**Lemma 4.3.** *Suppose Assumptions 4.1 and 4.2 hold. Then SAMPa satisfies the following descent inequality for $\rho > 0$ and a decreasing sequence $(\eta_t)_{t\in\mathbb{N}}$ with $\eta_t \in (0, \max\{1, c/L\})$ and $c \in (0, 1)$,*
$$\mathcal{V}_{t+1} \le \mathcal{V}_t - \eta_t(1 - \tfrac{\eta_t L}{2})\|\nabla f(x_t)\|^2 + \eta_t^2 \rho^2 C$$
*where $C = \frac{1}{2}(L^2 + L^3 + \frac{1}{1-c^2}L^4)$.*

Notice that $\eta_t$ is importantly squared in front of the error term $\rho^2 C$, while this is not the case for the term $-\|\nabla f(x_t)\|^2$. This allows us to control the error term while still providing convergence in terms of $\|\nabla f(x_t)\|^2$ as made precise by the following theorem.

**Theorem 4.4.** *Suppose Assumptions 4.1 and 4.2 hold. Then SAMPa satisfies the following descent inequality for $\rho > 0$ and a decreasing sequence $(\eta_t)_{t\in\mathbb{N}}$ with $\eta_t \in (0, \max\{1, 1/2L\})$,*
$$\sum_{t=0}^{T-1} \frac{\eta_t(1-\eta_t L/2)}{\sum_{\tau=0}^{T-1} \eta_\tau(1-\eta_\tau L/2)}\|\nabla f(x_t)\|^2 \le \frac{\Delta_0 + C\rho^2 \sum_{t=0}^{T-1} \eta_t^2}{\sum_{t=0}^{T-1} \eta_t(1-\eta_t L/2)} \tag{3}$$
*where $\Delta_0 = f(x_0) - \inf_{x\in\mathbb{R}^d} f(x)$ and $C = \frac{L^2+L^3}{2} + \frac{2L^4}{3}$. For $\eta_t = \min\{\frac{\sqrt{\Delta_0}}{\rho\sqrt{CT}}, \max\{\frac{1}{2L}, 1\}\}$*
$$\min_{t=0,\dots,T-1} \|\nabla f(x_t)\|^2 = \mathcal{O}\big(\frac{L\Delta_0}{T} + \frac{\rho\sqrt{\Delta_0 C}}{\sqrt{T}}\big) \tag{4}$$

*Remark* 4.5. Convergence follows as long as $\sum_{t=0}^{\infty} \eta_t = \infty$ and $\sum_{t=0}^{\infty} \eta_t^2 < \infty$, since the stepsize allows the right hand side to be made arbitrarily small. Note that Theorem 4.4 even allows for an *increasing* perturbation radius $\rho_t$, since it suffice to assume $\sum_{t=0}^{\infty} \eta_t = \infty$ and $\sum_{t=0}^{\infty} \eta_t^2 \rho_t^2 < \infty$.

## 5 Experiments

In this section, we demonstrate the benefit of SAMPa across a variety of models, datasets and tasks. It is worth noting that to enable parallel computation of SAMPa, we perform the two gradient calculations across 2 GPUs. As shown in Algorithm 1, one GPU computes $\nabla f(\widetilde{x}_t, \mathcal{B}_t)$ while another computes $\nabla f(y_{t+1}, \mathcal{B}_{t+1})$. For implementation guidance, we provide pseudo-code in Appendix E, along with algorithms detailing the integration of SAMPa with SGD and AdamW, both used as base optimizers in this section.

## 5.1 Image classification

**CIFAR-10/100.** We follow the experimental setup of Kwon et al. [2021]. We use the CIFAR-10 and CIFAR-100 datasets [Krizhevsky et al., 2009], both consisting of $50\,000$ training images of size $32 \times 32$, with 10 and 100 classes, respectively. For data augmentation, we apply the commonly used random cropping after padding with 4 pixels, horizontal flipping, and normalization using the statistics of the training distribution at both train and test time. We train multiple variants of VGG [Simonyan and Zisserman, 2014], ResNet [He et al., 2016], DenseNet [Huang et al., 2017] and WideResNet [Zagoruyko and Komodakis, 2016] (see Tables 1 and 2 for details) using cross entropy loss. All experiments are conducted on NVIDIA A100 GPU.

The models are trained using stochastic gradient descent (SGD) with a momentum of $0.9$ and a weight decay of $5 \times 10^{-4}$, both as a baseline and as the base model for SAM variants. We used a batch size of $128$ and a cosine learning rate schedule that starts at $0.1$. The number of epochs is set to 200 for SAM and SAMPa while SGD are given $400$ epochs. This is done in order to provide a computational fair comparison as SAM and SAMPa use twice as much gradient computation. Moreover, we show SAMPa-0.2 trained with $400$ epochs as a reference in the last column colored gray because SAMPa's theoretical limit of the per iteration cost is comparable to SGD. Note that all SAMPa-$\lambda$ in this section use the same number of epochs as SAM only except for the last column of Tables 1 and 2. Label smoothing with a factor of $0.1$ is employed for all methods.

Through a grid search over $\{0.01, 0.05, 0.1, 0.2, 0.4\}$ using the validation dataset on CIFAR-10 with ResNet-56, SAM is assigned $\rho$ values of $0.05$ and $0.1$ on CIFAR-10 and CIFAR-100 respectively, which is consistent with existing works [Foret et al., 2021, Kwon et al., 2021]. Moreover, SAMPa-0 shares the same $\rho$ value as SAM while SAMPa-0.2 is configured with twice the value of SAM's $\rho$. Additionally, $\lambda$ for SAMPa-$\lambda$ is set at $0.2$ through a grid search from $0$ to $1$, with intervals of $0.1$, with results detailed in Appendix B.3.

Training data is randomly partitioned into 90% for training and 10% for validation. To prevent overfitting on the test set, we deviate from Foret et al. [2021], Kwon et al. [2021] by selecting the model with the highest *validation* accuracy to report *test* accuracy. Results are averaged over 6 independent executions and presented in Tables 1 and 2. Compared with the enhancement from SGD to SAM, the average improvement from SAM to SAMPa-0.2 reaches 62.07% and 32.65% on CIFAR-10 and CIFAR-100, respectively.

Table 1: **Test accuracies on CIFAR-10.** SAMPa-0.2 outperforms SAM across all models with halved total temporal cost. "Temporal cost" represents the number of sequential gradient computations per update. SAMPa-0.2 with 400 epochs is included for comprehensive comparison with SGD and SAM.

| Model | SGD | SAM | SAMPa-0 | SAMPa-0.2 | SAMPa-0.2 |
|---|---|---|---|---|---|
| Temporal cost/Epochs | $\times 1/400$ | $\times 2/200$ | $\times 1/200$ | $\times 1/200$ | $\times 1/400$ |
| DenseNet-121 | $96.14_{\pm 0.09}$ | $96.49_{\pm 0.14}$ | $96.53_{\pm 0.11}$ | $\mathbf{96.77}_{\pm 0.11}$ | $96.92_{\pm 0.09}$ |
| Resnet-56 | $94.20_{\pm 0.39}$ | $94.26_{\pm 0.70}$ | $94.31_{\pm 0.43}$ | $\mathbf{94.62}_{\pm 0.35}$ | $95.43_{\pm 0.25}$ |
| VGG19-BN | $94.76_{\pm 0.10}$ | $95.05_{\pm 0.17}$ | $95.06_{\pm 0.22}$ | $\mathbf{95.11}_{\pm 0.10}$ | $95.34_{\pm 0.07}$ |
| WRN-28-2 | $95.71_{\pm 0.19}$ | $95.98_{\pm 0.10}$ | $96.06_{\pm 0.10}$ | $\mathbf{96.13}_{\pm 0.14}$ | $96.31_{\pm 0.09}$ |
| WRN-28-10 | $96.77_{\pm 0.21}$ | $97.25_{\pm 0.09}$ | $97.24_{\pm 0.11}$ | $\mathbf{97.34}_{\pm 0.09}$ | $97.46_{\pm 0.07}$ |
| Average | $95.52_{\pm 0.10}$ | $95.81_{\pm 0.15}$ | $95.86_{\pm 0.10}$ | $\mathbf{95.99}_{\pm 0.08}$ | $96.29_{\pm 0.06}$ |

Table 2: **Test accuracies on CIFAR-100.** SAMPa-0.2 outperforms SAM across all models with halved total temporal cost. "Temporal cost" represents the number of sequential gradient computations per update. SAMPa-0.2 with 400 epochs is included for a comprehensive comparison.

| Model | SGD | SAM | SAMPa-0 | SAMPa-0.2 | SAMPa-0.2 |
|---|---|---|---|---|---|
| Temporal cost/Epochs | $\times 1/400$ | $\times 2/200$ | $\times 1/200$ | $\times 1/200$ | $\times 1/400$ |
| DenseNet-121 | $81.08_{\pm 0.43}$ | $82.53_{\pm 0.22}$ | $82.50_{\pm 0.10}$ | $\mathbf{82.70}_{\pm 0.23}$ | $83.44_{\pm 0.21}$ |
| Resnet-56 | $74.09_{\pm 0.39}$ | $75.14_{\pm 0.15}$ | $75.22_{\pm 0.20}$ | $\mathbf{75.29}_{\pm 0.24}$ | $75.84_{\pm 0.27}$ |
| VGG19-BN | $74.85_{\pm 0.53}$ | $74.94_{\pm 0.12}$ | $74.94_{\pm 0.17}$ | $\mathbf{75.38}_{\pm 0.31}$ | $76.23_{\pm 0.16}$ |
| WRN-28-2 | $78.00_{\pm 0.17}$ | $78.50_{\pm 0.24}$ | $78.45_{\pm 0.29}$ | $\mathbf{78.82}_{\pm 0.22}$ | $79.46_{\pm 0.20}$ |
| WRN-28-10 | $81.56_{\pm 0.25}$ | $83.37_{\pm 0.30}$ | $83.46_{\pm 0.25}$ | $\mathbf{83.90}_{\pm 0.25}$ | $83.91_{\pm 0.13}$ |
| Average | $77.92_{\pm 0.17}$ | $78.90_{\pm 0.10}$ | $78.91_{\pm 0.09}$ | $\mathbf{79.22}_{\pm 0.11}$ | $79.78_{\pm 0.09}$ |

**ImageNet-1K.** We evaluate SAM and SAMPa-0.2 on ImageNet-1K [Russakovsky et al., 2015], using 90 training epochs, a weight decay of $10^{-4}$, and a batch size of 256. Other parameters match those of CIFAR-10. Each method undergoes 3 independent experiments, with test accuracies detailed in Table 3. Note that we omit SGD experiments due to computational constraints; however, prior research confirms SAM and its variants outperform SGD [Foret et al., 2021, Kwon et al., 2021].

Table 3: **Top1/Top5 maximum test accuracies on ImageNet-1K.**

|      | SAM                  | SAMPa-0.2            |
|------|----------------------|---------------------|
| Top1 | $77.25_{\pm 0.05}$   | $\mathbf{77.44}_{\pm 0.03}$ |
| Top5 | $93.60_{\pm 0.04}$   | $\mathbf{93.69}_{\pm 0.08}$ |

## 5.2 Efficiency comparison with efficient SAM variants

To comprehensively evaluate the efficiency gains of SAMPa compared to other variants of SAM in practical scenarios, we conduct experiments using five additional SAM variants on the CIFAR-10 dataset with ResNet-56 (detailed configuration in Appendix B.4): LookSAM [Liu et al., 2022], AE-SAM [Jiang et al., 2023], SAF [Du et al., 2022b], MESA [Du et al., 2022b], and ESAM [Du et al., 2022a]. Specifically, LookSAM alternates between SAM and a base optimizer periodically, while AE-SAM selectively employs SAM when detecting local sharpness. SAF and MESA eliminate the ascent step and introduce an extra trajectory loss term to reduce sharpness. ESAM leverages two strategies, *Stochastic Weight Perturbation (SWP)* and *Sharpness-sensitive Data Selection (SDS)*, for efficiency.

The number of sequentially computed gradients, as shown in Figure 2a, serves as a metric for computational time in an ideal scenario. Notably, SAMPa, SAF, and MESA require the fewest number of sequential gradients, each needing only half of SAM's. Specifically, SAF and MESA necessitate just one gradient computation per update, while SAMPa parallelizes two gradients per update.

However, real-world computational time encompasses more than just gradient computation; it includes forward and backward pass time, weight revision time, and potential communication overhead in distributed settings. Therefore, we present the actual training time in Figure 2b, revealing that SAMPa and SAF serve as the most efficient methods. LookSAM and AE-SAM, unable to entirely avoid computing two sequential gradients per update, exhibit greater time consumption than SAMPa as expected. MESA, requiring an additional forward step compared to the base optimizer during implementation, cannot halve the computation time relative to SAM's. Regarding ESAM, we solely integrate SWP in this experiment, as no efficiency advantage is observed compared to SAM when SDS is included. The reported time of SAMPa-0.2 in Figure 2b includes 7.5% communication overhead across GPUs. Achieving a nearly $2\times$ speedup in runtime could be possible with faster interconnects between GPUs. In addition, the test accuracies and the wall-clock time per epoch are reported in Table 4. SAMPa-0.2 achieves strong performance and meanwhile requires near-minimal computational time.

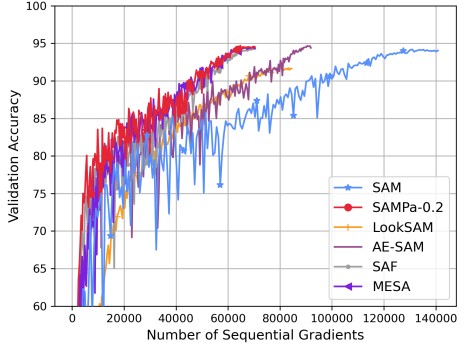

(a) Number of sequential gradients

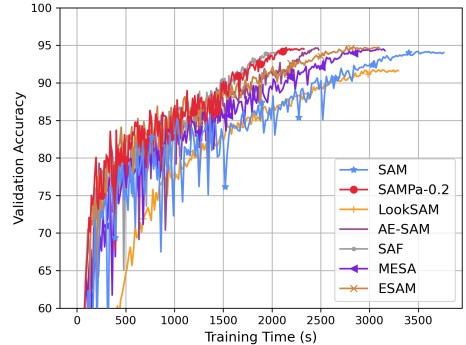

(b) Actual running time

Figure 2: **Computational time comparison for efficient SAM variants.** SAMPa-0.2 requires near-minimal computational time in both ideal and practical scenarios.

Table 4: **Efficient SAM variants.** The best result is in bold and the second best is underlined.

|                | SAM   | SAMPa-0.2       | LookSAM | AE-SAM        | SAF           | MESA  | ESAM  |
|----------------|-------|-----------------|---------|---------------|---------------|-------|-------|
| Accuracy       | 94.26 | **94.62**       | 91.42   | 94.46  | 93.89         | 94.23 | 94.21 |
| Time/Epoch (s) | 18.81 | 10.94    | 16.28   | 13.47         | **10.09**     | 15.43 | 15.97 |

## 5.3 Transfer learning

We demonstrate the benefits of SAMPa in transfer learning across vision and language domains.

**Image fine-tuning.** We conduct transfer learning experiments using the pre-trained ViT-B/16 checkpoint from Visual Transformers [Wu et al., 2020a], fine-tuning it on CIFAR-10 and CIFAR-100 datasets. AdamW is employed as the base optimizer, with gradient clipping applied at a global norm of 1. Training runs for 10 epochs, with a peak learning rate of $10^{-4}$. Other parameters remain consistent with those outlined in Section 5.1. Results in Table 5 show the benefits of SAMPa in image fine-tuning.

Table 5: **Image fine-tuning.** C10 and C100 represent CIFAR-10 and CIFAR-100 respectively.

|      | SAM | SAMPa-0.2 |
|------|-----|-----------|
| C10  | $98.87_{\pm 0.09}$ | $\mathbf{98.96}_{\pm 0.04}$ |
| C100 | $91.79_{\pm 0.12}$ | $\mathbf{93.06}_{\pm 0.16}$ |

**NLP fine-tuning.** To explore if SAMPa can benefit the natural language processing (NLP) domain, we show empirical text classification results in this section. In particular, we use BERT-base model and finetune it on the GLUE datasets [Wang et al., 2018]. We use AdamW as the base optimizer under a linear learning rate schedule and gradient clipping with global norm 1. We set the peak learning rate to $2 \times 10^{-5}$ and batch size to 32, and run 3 epochs with an exception for MRPC and WNLI which are significantly smaller datasets and where we used 5 epochs. Note that we set $\rho = 0.05$ for all datasets except for CoLA with $\rho = 0.01$, and RTE and STS-B with $\rho = 0.005$. The setting of $\rho$ is uniformly applied across SAM, SAMPa-0 and SAMPa-0.1. We report the results computed over 10 independent executions in the Table 6, which demonstrates that SAMPa also benefits in NLP domain.

Table 6: **Test results of BERT-base fine-tuned on GLUE.**

| Method | GLUE | CoLA | SST-2 | MRPC | STS-B | QQP | MNLI | QNLI | RTE | WNLI |
|--------|------|------|-------|------|-------|-----|------|------|-----|------|
|        |      | *Mcc.* | *Acc.* | *Acc./F1.* | *Pear./Spea.* | *Acc./F1.* | *Acc.* | *Acc.* | *Acc.* | *Acc.* |
| AdamW       | 74.6 | 56.6 | 91.6 | 85.6/89.9 | 85.4/85.3 | 90.2/86.8 | 82.6 | 89.8 | 62.4 | 26.4 |
| -w SAM      | 76.6 | 58.8 | 92.3 | 86.5/90.5 | 85.0/85.0 | 90.6/87.5 | 83.9 | 90.4 | 60.6 | 41.2 |
| -w SAMPa-0  | 76.9 | 58.9 | 92.5 | 86.4/90.4 | 85.0/85.0 | 90.6/87.6 | 83.8 | 90.4 | 60.4 | 43.2 |
| -w SAMPa-0.1 | **78.0** | 58.9 | 92.5 | 86.8/90.7 | 85.2/85.1 | 90.7/87.7 | 84.0 | 90.5 | 61.3 | 51.6 |

## 5.4 Noisy label task

We test on a task outside the i.i.d. setting that the method was designed for. Following Foret et al. [2021] we consider label noise, where a fraction of the labels in the training set are corrupted to another label sampled uniformly at random. Through a grid search over $\{0.005, 0.01, 0.05, 0.1, 0.2\}$, we set $\rho = 0.1$ for SAM, SAMPa-0 and SAMPa-0.2 except for adjusting $\rho = 0.01$ when the noise rate is $80\%$. Other experimental setup is the same as in Section 5.1. We find that SAMPa-0.2 enjoys better robustness to label noise than SAM.

Table 7: **Test accuracies of ResNet-32 models trained on CIFAR-10 with label noise.**

| Noise rate | SGD | SAM | SAMPa-0 | SAMPa-0.2 |
|------------|-----|-----|---------|-----------|
| 0%  | $94.22_{\pm 0.14}$ | $94.36_{\pm 0.07}$ | $94.36_{\pm 0.12}$ | $\mathbf{94.41}_{\pm 0.08}$ |
| 20% | $88.65_{\pm 0.75}$ | $92.20_{\pm 0.06}$ | $92.22_{\pm 0.10}$ | $\mathbf{92.39}_{\pm 0.09}$ |
| 40% | $84.24_{\pm 0.25}$ | $89.78_{\pm 0.12}$ | $89.75_{\pm 0.15}$ | $\mathbf{90.01}_{\pm 0.18}$ |
| 60% | $76.29_{\pm 0.25}$ | $83.83_{\pm 0.51}$ | $83.81_{\pm 0.37}$ | $\mathbf{84.38}_{\pm 0.07}$ |
| 80% | $44.44_{\pm 1.20}$ | $48.01_{\pm 1.63}$ | $48.22_{\pm 1.71}$ | $\mathbf{49.92}_{\pm 1.12}$ |

## 5.5 Incorporation with other SAM variants

We demonstrate the potential of SAMPa to enhance generalization further by integrating it with other variants of SAM. Specifically, we examine the results of combining SAMPa with five SAM variants: mSAM [Foret et al., 2021, Behdin et al., 2023], ASAM [Kwon et al., 2021], SAM-ON [Mueller et al., 2024], VaSSO [Li and Giannakis, 2024], and BiSAM [Xie et al., 2024]. Our experiments utilize Resnet-56 on CIFAR-10 trained with SAM and SAMPa-0.2, maintaining the same experimental setup as detailed in Section 5.1. Further specifics on the experimental configuration are provided in Appendix B.4. The results summarized in Table 8 underscore the seamless integration of SAMPa with these variants, leading to notable improvements in both generalization and efficiency.

Table 8: **Incorporation with variants of SAM.** SAMPa in the table denotes SAMPa-0.2. The incorporation of SAMPa with SAM variants enhances both accuracy and efficiency.

| mSAM | +SAMPa | ASAM | +SAMPa | SAM-ON | +SAMPa | VaSSO | +SAMPa | BiSAM | +SAMPa |
|---|---|---|---|---|---|---|---|---|---|
| 94.28 | **94.71** | 94.84 | **94.95** | 94.44 | **94.51** | 94.80 | **94.97** | 94.49 | **95.13** |

## 6 Related Works

**SAM.** Inspired by the strong correlation between the generalization of a model and the flat minima revealed in [Keskar et al., 2017, Jiang* et al., 2020], Foret et al. [2021] propose SAM seeking for a flat minima to improve generalization capability. SAM frames a minimax optimization problem that aims to achieve a minima whose neighborhoods also have low loss. To solve this minimax problem, the most popular way is using an ascent step to approximate the solution for the inner maximization problem with the fact that SAM with more ascent steps does not significantly enhance generalization [Kim et al., 2023]. Notably, SAM has demonstrated effectiveness across various supervised learning tasks in computer vision [Foret et al., 2021], with studies demonstrating the realm of NLP tasks [Bahri et al., 2021, Zhong et al., 2022].

**Efficient variants of SAM.** Compared with base optimizers like SGD, SAM doubles computational overhead stemming from its need for an extra gradient computation for perturbation per iteration. Efforts to alleviate SAM's computational burden have yielded several strategies. Firstly, strategies integrating SAM with base optimizers in an alternating fashion have been explored. For instance, *Randomized Sharpness-Aware Training (RST)* [Zhao et al., 2022b] employs a Bernoulli trial to randomly alternate between the base optimizer and SAM. Similarly, *LookSAM* [Liu et al., 2022] periodically computes the ascent step and utilizes the previous direction to promote flatness. Additionally, *Adaptive policy to Employ SAM (AE-SAM)* [Jiang et al., 2023] selectively applies SAM when detecting local sharpness, as indicated by the gradient norm.

Efficiency improvements have also been pursued by other means. *Efficient SAM (ESAM)* [Du et al., 2022a] enhances efficiency by leveraging less data, employing strategies such as Stochastic Weight Perturbation and Sharpness-sensitive Data Selection to subset random variables or mini-batch elements during optimization. Moreover, *Sparse SAM (SSAM)* [Mi et al., 2022] and *SAM-ON* [Mueller et al., 2024] achieve computational gains by only perturbing a subset of the model's weights, which enhances efficiency during the backward pass when only sparse gradients are needed. Notably, Du et al. [2022b] offer alternative approaches, *SAF* and *MESA*, estimating sharpness using loss trajectory instead of a single ascent step. Nonetheless, SAF requires increased memory consumption due to recording the outputs of historical models and MESA needs one extra forward pass. We compare against these methods in Section 5.2, where we find that SAMPa leads to a smaller wall-clock time.

## 7 Conclusion and Limitations

This paper introduces *Sharpness-aware Minimization Parallelized* (SAMPa) that halves the temporal cost of SAM through parallelizing gradient computations. The method additionally incorporates the optimistic gradient descent method. Crucially, SAMPa beats almost all existing efficient SAM variants regarding computational time in practice. Besides efficiency, numerical experiments demonstrate that SAMPa enhances the generalization among various tasks including image classification, transfer learning in vision and language domains, and noisy label tasks. SAMPa can be integrated with other SAM variants, offering both efficiency and generalization improvements. Furthermore, we show convergence guarantees for SAMPa even with a fixed perturbation size through a novel Lyapunov function, which we believe will benefit the development of SAM-based methods.

Although SAMPa achieves a $2\times$ speedup along with improved generalization, the computational resources required remain the same as SAM's, as two GPUs with equivalent memory (as discussed in Appendix D) are still needed. Future research could explore reducing costs by either: (i) eliminating the need for additional parallel computation, or (ii) reducing memory usage per GPU, making the resource requirements more affordable. Moreover, we prove convergence for SAMPa only in the specific case of $\lambda = 0$, leaving the analysis for general $\lambda$ as an open challenge for our future work.

## Acknowledgements

We thank the reviewers for their constructive feedback. This work was supported by the Swiss National Science Foundation (SNSF) under grant number 200021_205011. This work was supported by Google. This work was supported by Hasler Foundation Program: Hasler Responsible AI (project number 21043). This research was sponsored by the Army Research Office and was accomplished under Grant Number W911NF-24-1-0048.

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

# A    Proofs for Section 4

**Lemma 4.3.** *Suppose Assumptions 4.1 and 4.2 hold. Then SAMPa satisfies the following descent inequality for $\rho > 0$ and a decreasing sequence $(\eta_t)_{t \in \mathbb{N}}$ with $\eta_t \in (0, \max\{1, c/L\})$ and $c \in (0, 1)$,*

$$\mathcal{V}_{t+1} \leq \mathcal{V}_t - \eta_t(1 - \tfrac{\eta_t L}{2})\|\nabla f(x_t)\|^2 + \eta_t^2 \rho^2 C$$

*where $C = \frac{1}{2}(L^2 + L^3 + \frac{1}{1-c^2}L^4)$.*

*Proof.* Using smoothness we have

$$f(x_{t+1}) \leq f(x_t) + \langle \nabla f(x_t), x_{t+1} - x_t \rangle + \tfrac{L}{2}\|x_{t+1} - x_t\|^2$$

$$= f(x_t) - \eta_t \langle \nabla f(x_t), \nabla f(\widetilde{x}_t) \rangle + \tfrac{\eta_t^2 L}{2}\|\nabla f(\widetilde{x}_t)\|^2$$

$$= f(x_t) - \eta_t(1 - \tfrac{\eta_t L}{2})\|\nabla f(x_t)\|^2 + \tfrac{\eta_t^2 L}{2}\|\nabla f(\widetilde{x}_t) - \nabla f(x_t)\|^2$$

$$\quad - \eta_t(1 - \eta_t L) \langle \nabla f(x_t), \nabla f(\widetilde{x}_t) - \nabla f(x_t) \rangle$$

$$\text{(Assumption 4.2)} \leq f(x_t) - \eta_t(1 - \tfrac{\eta_t L}{2})\|\nabla f(x_t)\|^2 + \tfrac{\eta_t^2 \rho^2 L^3}{2}$$

$$\quad - \eta_t(1 - \eta_t L) \langle \nabla f(x_t), \nabla f(\widetilde{x}_t) - \nabla f(x_t) \rangle. \tag{5}$$

The last term of (5):

$$\langle \nabla f(x_t), \nabla f(\widetilde{x}_t) - \nabla f(x_t) \rangle = \langle \nabla f(x_t) - \nabla f(y_t) + \nabla f(y_t), \nabla f(\widetilde{x}_t) - \nabla f(x_t) \rangle$$

$$= \tfrac{\|\nabla f(y_t)\|}{\rho} \langle \widetilde{x}_t - x_t, \nabla f(\widetilde{x}_t) - \nabla f(x_t) \rangle$$

$$\quad + \langle \nabla f(x_t) - \nabla f(y_t), \nabla f(\widetilde{x}_t) - \nabla f(x_t) \rangle$$

$$\text{(Assumption 4.1)} \geq \langle \nabla f(x_t) - \nabla f(y_t), \nabla f(\widetilde{x}_t) - \nabla f(x_t) \rangle \tag{6}$$

For the left term, using $2 \langle a, \eta_t b \rangle = \|a\|^2 + \eta_t^2\|b\|^2 - \|a - \eta_t b\|^2$, we have

$$-2\eta_t \langle \nabla f(x_t) - \nabla f(y_t), \nabla f(\widetilde{x}_t) - \nabla f(x_t) \rangle$$

$$= 2 \langle \nabla f(x_t) - \nabla f(y_t), \eta_t(\nabla f(x_t) - \nabla f(\widetilde{x}_t)) \rangle$$

$$= -\|\nabla f(\widetilde{x}_t) - \nabla f(y_t) - (1 - \eta_t)(\nabla f(\widetilde{x}_t) - \nabla f(x_t))\|^2 + \|\nabla f(x_t) - \nabla f(y_t)\|^2$$

$$\quad + \eta_t^2\|\nabla f(\widetilde{x}_t) - \nabla f(x_t)\|^2$$

$$\leq -\tfrac{1}{1+e}\|\nabla f(\widetilde{x}_t) - \nabla f(y_t)\|^2 + (\eta_t^2 + \tfrac{(1-\eta_t)^2}{e})\|\nabla f(\widetilde{x}_t) - \nabla f(x_t)\|^2$$

$$\quad + \|\nabla f(x_t) - \nabla f(y_t)\|^2$$

$$\leq -\tfrac{1}{(1+e)\eta_t^2 L^2}\|\nabla f(x_{t+1}) - \nabla f(y_{t+1})\|^2 + (\eta_t^2 + \tfrac{(1-\eta_t)^2}{e})\|\nabla f(\widetilde{x}_t) - \nabla f(x_t)\|^2$$

$$\quad + \|\nabla f(x_t) - \nabla f(y_t)\|^2. \tag{7}$$

The first inequality is due to Young's inequality, $-\|a - b\|^2 \leq -\tfrac{1}{1+e}\|a\|^2 + \tfrac{1}{e}\|b\|^2$ with $e > 0$, while the second inequality follows from

$$\|\nabla f(y_t) - \nabla f(\widetilde{x}_t)\|^2 = \tfrac{1}{\eta_t^2}\|x_{t+1} - y_{t+1}\|^2 \geq \tfrac{1}{\eta_t^2 L^2}\|\nabla f(x_{t+1}) - \nabla f(y_{t+1})\|^2, \tag{8}$$

where the last inequality is due to Assumption 4.2. We can pick $e = \tfrac{1-\eta_t L}{\eta_t^2 L^2(1-\eta_{t+1}L)} - 1$ such that $\tfrac{1-\eta_t L}{(1+e)\eta_t^2 L^2} = 1 - \eta_{t+1}L$. To verify that $e > 0$, use that $(\eta_t)_{t \in \mathbb{N}}$ is decreasing to obtain

$$\tfrac{1-\eta_t L}{1-\eta_{t+1}L} \geq 1 \geq \eta_t^2 L^2 \tag{9}$$

where the last inequality uses that $\eta_t < 1/L$. Rearranging shows that $e > 0$.

With the particular choice of $e$, (7) reduces to

$$-2\eta_t \langle \nabla f(x_t) - \nabla f(y_t), \nabla f(\widetilde{x}_t) - \nabla f(x_t) \rangle$$

$$\leq -\tfrac{1-\eta_{t+1}L}{1-\eta_t L}\|\nabla f(x_{t+1}) - \nabla f(y_{t+1})\|^2 + \eta_t^2(1 + A_t)\|\nabla f(\widetilde{x}_t) - \nabla f(x_t)\|^2$$

$$\quad + \|\nabla f(x_t) - \nabla f(y_t)\|^2$$

$$\text{(Assumption 4.2)} \leq -\tfrac{1-\eta_{t+1}L}{1-\eta_t L}\|\nabla f(x_{t+1}) - \nabla f(y_{t+1})\|^2 + \eta_t^2(1 + A_t)\rho^2 L^2$$

$$\quad + \|\nabla f(x_t) - \nabla f(y_t)\|^2, \tag{10}$$

with $A_t = \frac{L^2(1-\eta_t)^2}{\frac{1-\eta_t L}{1-\eta_{t+1}L}-\eta_t^2 L^2}$. Plugging (6) and (10) back into (5) yields

$$f(x_{t+1}) + \tfrac{1}{2}(1-\eta_{t+1}L)\|\nabla f(x_{t+1}) - \nabla f(y_{t+1})\|^2$$
$$\leq f(x_t) + \tfrac{1}{2}(1-\eta_t L)\|\nabla f(x_t) - \nabla f(y_t)\|^2$$
$$- \eta_t(1 - \tfrac{\eta_t L}{2})\|\nabla f(x_t)\|^2 + \tfrac{1}{2}\eta_t^2\Big((1-\eta_t L)(1+A_t) + L\Big)L^2\rho^2 \qquad (11)$$

What remains is to bound the latter term of (11) in terms of a constant independent of $t$. First notice that, due to the first inequality of (9) and $\eta_t L < c$ by assumption, we have

$$\frac{1-\eta_t L}{1-\eta_{t+1}L} - \eta_t^2 L^2 > 1 - c^2.$$

It follows that

$$\frac{(1-\eta_t L)(1-\eta_t)^2}{\frac{1-\eta_t L}{1-\eta_{t+1}L}-\eta_t^2 L^2} < \frac{(1-\eta_t L)(1-\eta_t)^2}{1-c^2} < \frac{1}{1-c^2} \qquad (12)$$

where the last inequality uses $\eta_t < \frac{1}{L}$ and $\eta_t \leq 1$.

Expanding the last term of (11), we obtain

$$\tfrac{1}{2}\big((1-\eta_t L)(1+A_t) + L\big)L^2 = \tfrac{1}{2}\Big(1 - \eta_t L + L^2\frac{(1-\eta_t L)(1-\eta_t)^2}{\frac{1-\eta_t L}{1-\eta_{t+1}L}-\eta_t^2 L^2} + L\Big)L^2$$
$$\leq \tfrac{1}{2}\Big(1 + L^2\frac{(1-\eta_t L)(1-\eta_t)^2}{\frac{1-\eta_t L}{1-\eta_{t+1}L}-\eta_t^2 L^2} + L\Big)L^2$$
$$(12) \leq \frac{1 + L + \frac{1}{1-c^2}L^2}{2}L^2 =: C,$$

which completes the proof. $\qquad\square$

**Theorem 4.4.** *Suppose Assumptions 4.1 and 4.2 hold. Then SAMPa satisfies the following descent inequality for $\rho > 0$ and a decreasing sequence $(\eta_t)_{t\in\mathbb{N}}$ with $\eta_t \in (0, \max\{1, 1/2L\})$,*

$$\sum_{t=0}^{T-1}\frac{\eta_t(1-\eta_t L/2)}{\sum_{\tau=0}^{T-1}\eta_\tau(1-\eta_\tau L/2)}\|\nabla f(x_t)\|^2 \leq \frac{\Delta_0 + C\rho^2\sum_{t=0}^{T-1}\eta_t^2}{\sum_{t=0}^{T-1}\eta_t(1-\eta_t L/2)} \qquad (3)$$

*where $\Delta_0 = f(x_0) - \inf_{x\in\mathbb{R}^d} f(x)$ and $C = \frac{L^2+L^3}{2} + \frac{2L^4}{3}$. For $\eta_t = \min\{\frac{\sqrt{\Delta_0}}{\rho\sqrt{CT}}, \max\{\frac{1}{2L}, 1\}\}$*

$$\min_{t=0,\dots,T-1}\|\nabla f(x_t)\|^2 = \mathcal{O}\big(\tfrac{L\Delta_0}{T} + \tfrac{\rho\sqrt{\Delta_0 C}}{\sqrt{T}}\big) \qquad (4)$$

*Proof.* The proof follows directly by telescoping the descent inequality from Lemma 4.3 after subtracting $\inf_{x\in\mathbb{R}^d} f(x)$ from which we have

$$\sum_{t=0}^{T-1}\frac{\eta_t(1-\eta_t L/2)}{\sum_{\tau=0}^{T-1}\eta_\tau(1-\eta_\tau L/2)}\|\nabla f(x_t)\|^2 \leq \frac{\Delta_0 + \frac{1}{2}\|\nabla f(x_0) - \nabla f(y_0)\|^2 + C\rho^2\sum_{t=0}^{T-1}\eta_t^2}{\sum_{t=0}^{T-1}\eta_t(1-\eta_t L/2)} \qquad (13)$$

Using Lipschitz continuity from Assumption 4.2 we have that

$$\|\nabla f(x_0) - \nabla f(y_0)\|^2 \leq L^2\|x_0 - y_0\|^2 = 0 \qquad (14)$$

where the last equality follows from picking the initialization $y_0 = x_0$. By picking $c = \frac{1}{2}$, for which $C$ simplifies and $\eta_t < \frac{1}{2L}$, we obtain the guarantee in (3).

Picking a fixed stepsize $\eta_t = \eta$, the convergence guarantee (3) reduces to

$$\tfrac{1}{T}\sum_{t=0}^{T-1}\|\nabla f(x_t)\|^2 \leq \tfrac{4}{3}\big(\tfrac{\Delta_0}{T\eta} + C\rho^2\eta\big) \qquad (15)$$

Optimizing the bound suggests a stepsize of $\sqrt{\frac{\Delta_0}{C\rho^2 T}}$. Thus, incorporating the other stepsize requirements, we set $\eta = \min\{\sqrt{\frac{\Delta_0}{C\rho^2 T}}, \max\{\frac{1}{2L}, 1\}\}$. There are three cases.

**Case I** $\eta = \sqrt{\frac{\Delta_0}{C\rho^2 T}}$ for which (15) reduces to

$$\tfrac{1}{T}\sum_{t=0}^{T-1}\|\nabla f(x_t)\|^2 \leq \tfrac{8}{3}\frac{\rho\sqrt{\Delta_0 C}}{\sqrt{T}} \qquad (16)$$

**Case II** $\eta = \frac{1}{2L} \leq \sqrt{\frac{\Delta_0}{C\rho^2 T}}$ for which

$$\frac{1}{T} \sum_{t=0}^{T-1} \|\nabla f(x_t)\|^2 \leq \frac{4}{3}\left(\frac{2L\Delta_0}{T} + \frac{\rho\sqrt{\Delta_0 C}}{\sqrt{T}}\right) \tag{17}$$

**Case III** $\eta = 1 \leq \sqrt{\frac{\Delta_0}{C\rho^2 T}}$ we can additionally use that $1 \geq \frac{1}{2L}$, to again establish (17).

Combining the three cases, we have that for any case

$$\frac{1}{T} \sum_{t=0}^{T-1} \|\nabla f(x_t)\|^2 = \mathcal{O}\left(\frac{L\Delta_0}{T} + \frac{\rho\sqrt{\Delta_0 C}}{\sqrt{T}}\right) \tag{18}$$

Noting that the minimum is always smaller than the average completes the proof. $\qquad\square$

# B  Additional Experiments

## B.1  Failure of OptSAM

To demonstrate the failure of our naive attempt, described in Section 2.2 as OptSAM, we provide empirical results using identical settings in Section 5.1. As shown in Table 9, OptSAM *fails* to outperform SAM and even performs worse than SGD.

Table 9: Test accuracies of OptSAM on CIFAR-10.

| Model | SGD | SAM | OptSAM |
|---|---|---|---|
| Resnet-56 | 94.20 | 94.26 | 93.99 |
| WRN-28-2 | 95.71 | 95.98 | 95.41 |
| VGG19-BN | 94.76 | 95.05 | 94.32 |

## B.2  SAM with stochasticity

To deploy SAM with stochasticity, we find it imperative to utilize the same batch for both gradient calculations of perturbation and correction steps. Otherwise, the performance of SAM may be even worse than the base optimizer. Our empirical observations on CIFAR-10 are shown in Table 10, which is also validated by [Li and Giannakis, 2024, Li et al., 2024].

This observation demonstrates that the same batch for both perturbation and correction steps is essential. This also justifies the need for parallel gradient computation on *two sequential batches* in SAMPa.

Table 10: **Two gradients in SAM computed on the same or different batch on CIFAR-10.** SAM computes them on the same batch while SAM-db is on two different batches.

| Model | SGD | SAM | SAM-db |
|---|---|---|---|
| Resnet-56 | 94.20 | 94.26 | 93.97 |
| WRN-28-2 | 95.71 | 95.98 | 95.50 |
| VGG19-BN | 94.76 | 95.05 | 94.48 |

## B.3  Sweep over $\lambda$ for SAMPa

To investigate the impact of different values of $\lambda$ in SAMPa, we present test accuracy curves for ResNet-56 and WRN-28-10 on CIFAR-10 in Figure 3, covering the range $\lambda \in [0, 1]$ with the interval 0.1. Notably, SAMPa-1 corresponds to OptGD.

In our experiments, as reported in Table 1 and Table 2, we initially optimize $\lambda = 0.2$ using ResNet-56. This default value is applied consistently across other models to maintain a fair comparison. However, continuous tuning of $\lambda$ may lead to improved performance on different model architectures, as demonstrated in Figure 3b.

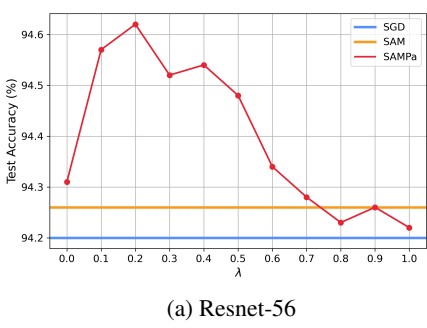
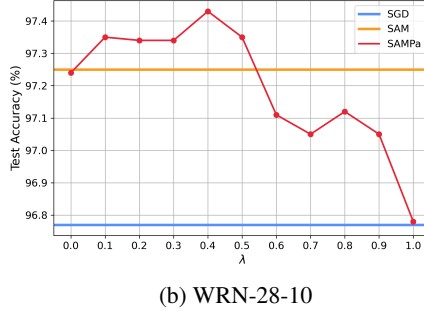

(a) Resnet-56                                    (b) WRN-28-10

Figure 3: Test accuracy curve obtained from SAMPa algorithm using a range of $\lambda$.

## B.4 Hyperparameters for variants of SAM

We present a comparison between SAMPa and various variants of SAM in Table 4 and Table 8. All algorithms in these tables utilize Resnet-56 on CIFAR-10 with hyperparameters mostly consistent with those used in Table 1. However, some variants require additional or different hyperparameters, which are listed below:

- LookSAM [Liu et al., 2022]: Update frequency $k = 5$, scaling factor $\alpha = 0.7$.
- AE-SAM [Jiang et al., 2023]: $\rho = 0.2$, forgetting rate $\delta = 0.9$.
- MESA [Du et al., 2022b]: Starting epoch $E_{\text{start}} = 5$, coefficients $\lambda = 0.8$, Decay factor $\beta = 0.9995$.
- ESAM [Du et al., 2022a]: SWP probability $\beta = 0.5$.
- mSAM [Foret et al., 2021]: Size of micro batch $m = 32$.
- ASAM [Kwon et al., 2021]: $\rho = 0.5$.
- SAM-ON [Mueller et al., 2024]: $\rho = 0.5$.
- VaSSO [Li and Giannakis, 2024]: Linearization parameter $\theta = 0.4$.
- BiSAM Xie et al. [2024]: BiSAM (-log) with $\mu = 1$.

We adhere to the default values specified in the original papers and maintain consistent naming conventions. Following the experimental setup detailed in Section 5.1, we set $\rho \times 2$ for SAMPa in Section 5.5 when incorporated with the algorithms, while keeping other parameters consistent with their defaults. Note that these parameters are not tuned to ensure a fair comparison and avoid waste of computing resources.

## B.5 mSAM with 2 GPUs

Since SAMPa parallelizes two gradient computations across two GPUs, we implement mSAM [Behdin et al., 2023], a SAM variant that achieves data parallelism, for a fair comparison of runtime. Based on experiments in Section 5.2, mSAM (m=2) uses two GPUs and each computes gradient for 64 samples. While mSAM's total computation time for batch sizes of 64 and 128 is similar, its wall-clock time is slightly longer due to the added communication overhead between GPUs. This highlights the need for gradient parallelization.

Table 11: Runtime of SAM variants on 2 GPUs.

|  | SAM | mSAM (m=2) | SAMPa-0.2 |
|---|---|---|---|
| Number of GPUs | 1 | 2 | 2 |
| Time/Epoch (s) | 18.81 | 21.17 | 10.94 |

We also provide the runtime per batch of SGD across various batch sizes in Table 12. The results show that data parallelism reduces time efficiently only when the batch size is sufficiently large. However, excessively large batch sizes can negatively affect generalization [He et al., 2019].

Table 12: Runtime per batch/epoch of different batch sizes.

| Batch size | 64 | 128 | 256 | 512 | 1024 | 2048 | 4096 |
|---|---|---|---|---|---|---|---|
| Time/Batch (ms) | 21.70 | 22.70 | 23.08 | 27.84 | 32.88 | 50.00 | 120.16 |

### B.6 SAMPa-$\lambda$ v.s. the gradient penalization method

SAMPa-$\lambda$ takes a convex combination of the two convergent schemes $x_{t+1} = (1 - \lambda)\,\mathrm{SAMPa}(x_t) + \lambda\,\mathrm{OptGD}(x_t)$, which is similar with a gradient penalization method [Zhao et al., 2022a] doing $x_{t+1} = (1 - \lambda)\,\mathrm{SAM}(x_t) + \lambda\,\mathrm{SGD}(x_t)$. However, it is important to note that SAMPa-$\lambda$ differs in a key aspect: it computes gradients for each update on two different batches (as shown in line 6 of Algorithm 1), while the penalizing method combines gradients from the same batch.

We conducted preliminary experiments on CIFAR-10 using the penalizing method with the same hyperparameters as SAMPa-0.2. The results indicate similar performance in standard classification tasks but show worse outcomes with noisy labels. Further investigation into this discrepancy may provide insights into SAMPa's superior performance.

Table 13: Test accuracy of the gradient penalization method.

| | SAM | SAMPa-0.2 | Penalizing |
|---|---|---|---|
| Resnet-56 | 94.26 | 94.62 | 94.57 |
| Resnet-32 (80% noisy label) | 48.01 | 49.92 | 48.26 |

## C   The choice of $y_{t+1}$

The particular choice of $y_{t+1}$ in SAMPa is a direct consequence of the analysis. Specifically, in Equation (8) of the proof, the choice $y_{t+1} = x_t - \eta_t \nabla f(y_t)$ allows us to produce the term $\|\nabla f(x_{t+1}) - \nabla f(y_{t+1})\|^2$ in order to telescope with $\|\nabla f(x_t) - \nabla f(y_t)\|^2$ in Equation (7). This is what we refer to in Section 3, when mentioning that we will pick $y_t$ such that $\|\nabla f(x_t) - \nabla f(y_t)\|^2$ (i.e. the discrepancy from SAM) can be controlled. This gives a precise guarantee explaining why $\nabla f(x_t)$ can be replaced by $\nabla f(y_t)$.

Additionally, the small difference between the perturbations based on $\nabla f(x_t)$ and $\nabla f(y_t)$ suggests that $\nabla f(y_t)$ serves as an effective approximation of $\nabla f(x_t)$ in practice. In Figure 4, we track the cosine similarity and Euclidean distance between $\nabla f(y_t)$ and $\nabla f(x_t)$ throughout the training process of ResNet-56 on CIFAR-10 . We find that the cosine similarity keeps above 0.99 during the whole training process, and in most period it's around 0.998, while at the end of training it is even close to 1. This indicates that SAMPa's estimated perturbation is an excellent approximation of SAM's perturbation.

Moreover, the Euclidean distance decreases and is close to zero at the end of training. This matches our theoretical analysis that $\|\nabla f(x_t) - \nabla f(y_t)\|^2$ eventually becomes small, which lemma 4.3 guarantees in the convex case by establishing the decrease of the potential function $\mathcal{V}_t$.

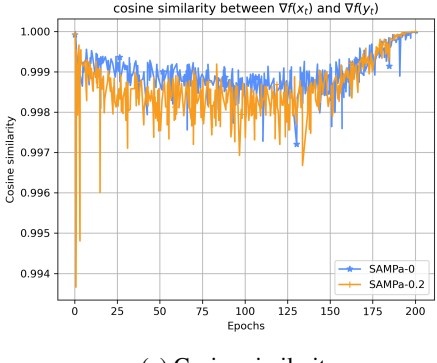

(a) Cosine similarity

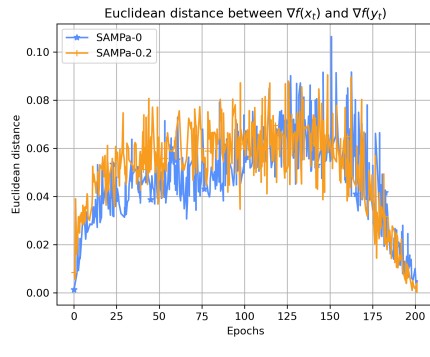

(b) Euclidean distance

Figure 4: Difference between $\nabla f(x_t)$ and $\nabla f(y_t)$.

# D Discussion of memory usage

From the implementation perspective, it is worth discussing the memory usage of SAMPa compared with SAM. As depicted in SAM, SAM necessitates the storage of a maximum of two sets of model weights $(x_t, \tilde{x}_t)$, along with one gradient $(\nabla f(x_t))$, and one mini-batch $(\mathcal{B}_t)$ for each update. Benefiting from 2 GPUs' deployment, SAMPa-$\lambda$ requires the same memory usage as SAM on *each* GPU, specifically needing two model weights $(x_t, \tilde{x}_t$ or $y_{t+1})$, one gradient $(\nabla f(x_t)$ or $\nabla f(y_{t+1}))$, and one mini-batch $(\mathcal{B}_t$ or $\mathcal{B}_{t+1})$.

We present a memory usage comparison in Table 14 for all SAM variants introduced in Section 5.2. Notably, SAMPa-0.2 requires slightly less memory per GPU, while MESA consumes approximately 23% more memory than SAM. The other three methods have comparable memory usage to SAM. However, it's important to note that memory usage is highly dependent on the size of the model and dataset, particularly for SAF and MESA, which store historical model outputs or weights.

Table 14: Memory usage on each GPU.

|  | SAM | SAMPa-0.2 | LookSAM | AE-SAM | SAF | MESA | ESAM |
|---|---|---|---|---|---|---|---|
| Memory (MiB) | 2290 | 2016 | 2296 | 2292 | 2294 | 2814 | 2288 |

# E Implementation guidelines

Our algorithm SAMPa is deployed across two GPUs to facilitate parallel training. As shown in Algorithm 1, one GPU calculates $\nabla f(\tilde{x}_t, \mathcal{B}_t)$ and another one takes responsibility for $\nabla f(y_{t+1}, \mathcal{B}_{t+1})$. For ease of implementation, we provide a detailed version in Algorithm 2, with the following key points:

- Apart from the synchronization step (line 8), all other operations can be executed in parallel on both GPUs.
- The optimizer state, $m$, used in $\mathrm{ModelUpdate}_m()$ includes necessary elements such as step size, momentum, and weight decay. Crucially, to ensure that $y_{t+1}$ (line 6) is close to $x_{t+1}$, the update for $y_{t+1}$ uses $m_t$, the state associated with $x_t$. Note that the optimizer state is not updated in line 6.

---

**Algorithm 2** SAMPa on two GPUs

---

**Input:** Initialization $x_0 \in \mathbb{R}^d$, initialization $y_0 = x_0$ and $g_0 = \nabla f(y_0, \mathcal{B}_0)$, iterations $T$, step sizes $\{\eta_t\}_{t=0}^{T-1}$, neighborhood size $\rho > 0$, interpolation ratio $\lambda$, optimizer state $m_0$.

1 **for** $t = 0$ **to** $T - 1$ **do**
2      **GPU1:** Load minibatch $\mathcal{B}_t$.
3      **GPU1:** Compute perturbed weight $\tilde{x}_t = x_t + \rho \frac{g_t}{\|g_t\|}$.
4      **GPU1:** Compute gradient $\tilde{g}_t = \nabla f(\tilde{x}_t, \mathcal{B}_t)$.
5      **GPU2:** Load minibatch $\mathcal{B}_{t+1}$.
6      **GPU2:** Compute the auxiliary sequence $y_{t+1}, \_ = \mathrm{ModelUpdate}_{m_t}(x_t, g_t)$.
7      **GPU2:** Compute gradient $g_{t+1} = \nabla f(y_{t+1}, \mathcal{B}_{t+1})$.
8      **Both:** Communicate $\tilde{g}_t$ and $g_{t+1}$ between GPU1 and GPU2.      ▷ Synchronization barrier
9      **Both:** Compute the final gradient $G_t = (1 - \lambda)\tilde{g}_t + \lambda g_{t+1}$.
10      **Both:** Update weights $x_{t+1}, m_{t+1} = \mathrm{ModelUpdate}_{m_t}(x_t, G_t)$.      ▷ Updates optimizer state

---

