# OpenReview forum: "SAMPa: Sharpness-aware Minimization Parallelized"
_NeurIPS.cc/2024/Conference — NeurIPS 2024 poster_

### Official Review · Reviewer_9hZ5 · 2024-06-19

**Soundness:** 2
**Presentation:** 3
**Contribution:** 2
**Rating:** 5
**Confidence:** 4

**Summary:**

This paper studies the efficiency problem of sharpness-aware minimization (SAM) algorithms.
SAM requires two gradient calculations: one for computing the perturbation and another for computing the update direction.
Hence, SAM doubles the computation cost compared with ERM.
Furthermore, these two gradients cannot be parallelized.
The authors introduce an auxiliary sequence and use the gradient on the auxiliary variable to approximate the gradient in computing the perturbation.
Experiments are conducted to evaluate the effectiveness of the proposed SAMPa algorithm.

**Strengths:**

- improving the efficiency of SAM is an important problem
- moreover, making the two gradient computations parallelized is crucial to accelerate SAM algorithms
- introducing an auxiliary sequence to approximate the gradient in computing perturbation is interesting
- the writing is clear and easy to follow.

**Weaknesses:**

- Though introducing an auxiliary sequence is an interesting idea to parallelize the two gradient calculations,
I have a concern about the auxiliary sequence:
    -  estimate the approximate error $\nabla f(x_t, B_{t+1}) - \nabla f(y_{t+1}, B_{t+1}) $. Establishing a bound for this error is crucial to the proposed algorithm. Otherwise, the method has no theoretical support.

- The theoretical analysis in Section 4 requires the loss to be convex.
However, losses are usually non-convex in practice (for example, experiments in the paper).
Thus, the analysis in section 4 is meaningless in practice.

- Main results (Tables 1 and 2) show that SAMPa outperforms SAM in all settings.
This observation is odd to me.
Note that SAMPa uses an approximate method to compute the gradient but achieves better performance than SAM which uses the exact gradient.
Some explanations are needed here.

**Questions:**

see above weaknesses.

---

> ### Author Rebuttal · Authors · 2024-08-07
>
> We thank the reviewer for their valuable feedback and address all remaining concerns below:
>
> > Q1. Though introducing an auxiliary sequence is an interesting idea to parallelize the two gradient calculations, I have a concern about the auxiliary sequence:
> Estimate the approximate error $\nabla f(x_t, \mathcal B_{t+1}) - \nabla f(y_{t+1}, \mathcal B_{t+1})$. Establishing a bound for this error is crucial to the proposed algorithm. Otherwise, the method has no theoretical support.
>
> A1. In the proof we do not actually need a bound on $||\nabla f(x_t) - \nabla f(y_{t})||^2$, since we can telescope the quantity and entirely remove it from the resulting rate by choice of the initialization (see l. 165-168). The only thing we need is that the quantity eventually becomes small, which Lemma 4.3 guarantees in the convex case by establishing decrease of the potential function $\mathcal V_t$.
>
> In practice we observe that the difference between $\nabla f(x_t, \mathcal B_t)$ and $\nabla f(y_t, \mathcal B_t)$ is in fact small. Specifically, we track the cosine similarity and Euclidean distance between $\nabla f(x_t, \mathcal B_t)$ and $\nabla f(y_t, \mathcal B_t)$ during the whole training process of Resnet-56 on CIFAR-10 [here](https://imgur.com/a/6hWQm44). We find that the cosine similarity keeps above 0.99 during the whole training process, and in most periods it is around 0.998, while at the end of training it is even close to 1. This indicates that SAMPa's approximated perturbation $\rho \frac{\nabla f(y_t)}{||\nabla f(y_t)||}$ is an excellent approximation of SAM's perturbation $\rho \frac{\nabla f(x_t)}{||\nabla f(x_t)||}$.
>
> Moreover, the Euclidean distance decreases and is close to zero at the end of training. This matches our above theoretical analysis that $||\nabla f(x_t) -\nabla f(y_t)||^2$ eventually becomes small.
>
> > Q2. The theoretical analysis in Section 4 requires the loss to be convex. However, losses are usually non-convex in practice (for example, experiments in the paper). Thus, the analysis in section 4 is meaningless in practice.
>
> A2. We kindly disagree that convex analysis is "meaningless" in guiding nonconvex practice. Arguably most of the successful optimizers today for deep learning were developed under the assumption of convexity. This includes AdaGrad, Adam and even more recent developments such as Distributed Shampoo and the Scheduler-free optimizer that won the AlgoPerf competition this month [1].
>
> Similarly, we found convex analysis extremely helpful in developing the algorithm. In our particular case, our algorithmic design _falls out of the analysis_, specifically leading to our choice of the auxilliary sequence $(y_t)_{t\in \mathbb N}$ (see Section 3).
>
> We subsequently show that SAMPa indeed has strong practical performance in deep learning applications (see Section 5). As mentioned in Q1, we additionally provide evidence that $y_t$ indeed acts as a good estimator for $x_t$ in the experiments, which builds further confidence in the approach.
>
> > Q3. Main results (Tables 1 and 2) show that SAMPa outperforms SAM in all settings. This observation is odd to me. Note that SAMPa uses an approximate method to compute the gradient but achieves better performance than SAM which uses the exact gradient. Some explanations are needed here.
>
> A3. There is no conflict between the two observations:
>
> - Our theoretical claim arguing through closeness to SAM is concerned with SAMPa-0 (referred to as simply SAMPa in line 128). SAM and SAMPa-0 have very similar performance in practice, which in fact _confirms_ the theory (see Table 1, 2, 6 and 7).
> - The method that consistently beats SAM is SAMPa-0.2, which is simply a convex combination of SAMPa and OptGD (see l. 138 for details). It is an interesting direction to understand this superior performance better. One hypothesis for the superior performance is that optimistic gradient descent might be better suited for nonsmooth parts of the optimization landscape and thus stabilize the algorithm when Lipschitz continuity assumption is not present.
>
> We would like to highlight again that SAMPa-0.2 _consistently_ outperforms SAM with 2$\times$ speedup across models and datasets (see Section 5). It can also be run on a single device with the same runtime as SAM, while _still delivering the same superior performance_. We will remark on this in the final version.
>
>
> [1] https://mlcommons.org/2024/08/mlc-algoperf-benchmark-competition/

---

> ### Comment · Reviewer_9hZ5 · 2024-08-13
> **reply to rebuttal**
>
> Thanks for the authors' reply.
>
> Indeed, I mean a bound on the approximate error $\nabla f(x_t, B_{t+1}) - \nabla f(y_{t+1}, B_{t+1}) $ in the **non-convex** case!
>
> For Q2, As SAM is designed for non-convex problems, I do not think it is meaningful to focus on analyzing the convergence in the convex case.

---

> > ### Author Response · Authors · 2024-08-14
> > **Further response to Reviewer 9hZ5**
> >
> > We sincerely thank the reviewer for the positive response.
> >
> > Regarding the focus on convex analysis, we do not understand why convex analysis should be less suitable for SAM than e.g. the mentioned AdaGrad or Adam. SAM has mostly seen success in overparameterized settings (where the training data is eventually perfectly fitted), where it has been exploited in several works that a convexification can occur. Additionally, the solution set for the convex problem is not necessarily a singleton, so biasing the obtained solution still seems reasonable.
> >
> > Moreover, we do not see a bound on the approximate error as crucial, since it is not our primary goal. At least in the convex case, convergence can be established _without_ such a bound as we demonstrated in the rebuttal. We are unsure why it would become essential in the non-convex setting.
> >
> > We would like to explore the convergence in the non-convex case in our future work. Thank you again for your engagement, and we remain available for any further questions.

---

### Official Review · Reviewer_LruU · 2024-06-28

**Soundness:** 3
**Presentation:** 3
**Contribution:** 2
**Rating:** 6
**Confidence:** 4

**Summary:**

This paper studies a parallelized variant of sharpness aware minimization (SAM). This is achieved by introducing a sequence of auxiliary variables to break down the sequential dependence for the two gradients in every SAM iteration. The resultant approach SAMPa has convergence guarantee for convex problems when $\rho$ is chosen as a constant independent of $T$. A heuristic (SAMPa-$\lambda$) combining SAMPa with optimistic gradient descent method is also proposed. Numerical results show that SAMPa and SAMPa-$\lambda$ generalization merits and runtime benefits.

**Strengths:**

S1. **Novel methodology.** Using a new sequence to break the sequential dependence of SAM gradients is novel.

S2. **Flexible framework and improved test performance.** As shown in Section 5.5, the proposed approach can be integrated with different variants of SAM. Moreover, the proposed approach seems to improve the test performance on various downstream tasks.

**Weaknesses:**

W1. My major concern is that the runtime comparison is not fair due to the second GPU. In particular, since SAMPa uses 2 GPU, can the authors also report data parallel for SAM with 2GPUs as well? (or m-SAM for communication efficiency). For example, if the batch size for SAM is 128, each GPU can calculate gradients for 64 samples in parallel.

W2. While theoretical results for deterministic convex functions are provided, the metric is gradient norm. It is known that gradient norm is easier to be optimized compared with function values [1]. Can the authors comment more on the optimality of this convergence rate? Does SAMPa-$\lambda$ also enjoy theoretical guarantees?

W3. More on parallelization should be discussed.
For example, how does the memory consumption of SAMPa, and how does it compare to other efficient SAM variants? How is the communication of SAMPa? Can communication be overlapped with computation? These questions should also be discussed in detail as the parallelization in SAMPa is the key novelty.



[1] Allen-Zhu, Zeyuan. "How to make the gradients small stochastically: Even faster convex and nonconvex sgd." Advances in Neural Information Processing Systems 31 (2018).

**Questions:**

Q1. In algorithm 1, is $g_0=\nabla f(y_0)$ or $g_0=\nabla f(y_0, {\cal B}_0)$?

Q2. Can the authors comment more on the intuition on the reason that SAMPa outperforms SAM?


There are also typos:
- line 180, we can trivially allows --> we can trivially allow
- line 126, identical to SAM except .. --> consider to change the word 'identical'

**Limitations:**

The authors have discussed the limitations.

---

> ### Author Rebuttal · Authors · 2024-08-07
>
> We thank the reviewer for their valuable feedback and address all remaining concerns below:
>
> > Q1.  My major concern is that the runtime comparison is not fair due to the second GPU. In particular, since SAMPa uses 2 GPUs, can the authors also report data parallel for SAM with 2 GPUs as well? (or mSAM for communication efficiency). For example, if the batch size for SAM is 128, each GPU can calculate gradients for 64 samples in parallel.
>
> A1. We report the runtime for mSAM with 2 GPUs as suggested. Each GPU calculates gradients for 64 samples in parallel, based on experiments in Section 5.2. mSAM requires slightly longer wall-clock time due to similar computation times for batch sizes of 64 and 128, but with added communication overhead. This highlights the need for gradient parallelization. Moreover, SAMPa-0.2 can run on a single device with the same runtime as SAM, yet achieves superior performance.
>
> |   | SAM | mSAM (m=2) | SAMPa-0.2 |
> |:---:|:---:|:---:|:---:|
> | Time/Epoch (s) |  18.81  | 22.43 | 10.94 |
>
> We additionally provide the time per batch and per epoch for SGD with different batch sizes, showing that doubling the batch size slightly increases computation time per batch, while the total epoch time decreases.
>
> | Batch size  | 64 | 128 | 512 | 2048 |
> |:---:|:---:|:---:|:---:|:---:|
> | Time/Batch (ms) | 21.70 | 22.70 | 27.84 | 50.00 |
> | Time/Epoch (s) | 15.28 | 7.99 | 2.45 | 1.10 |
>
> > Q2.1 Can the authors comment more on the optimality of this convergence rate?
>
> A2.1. The rate matches those of SAM (see Thm. 3.3 Si and Yun [2024]) and recovers the rate of gradient descent when $\rho=0$, but they are not optimal for the given problem-class for which acceleration is possible.
>
> However, note that our goal is not to establish optimal rates, but we rather use the analysis to directly construct our algorithm, by requiring the guarantee to match that of SAM. Specifically in Eq. 7 of the proof, the choice of the auxiliary sequence $y_{t+1} = x_t - \eta_t \nabla f(y_t)$ is what allows us to produce the term $||\nabla f(x_{t+1})-\nabla f(y_{t+1})||^2$ in order to telescope with $||\nabla f(x_{t})-\nabla f(y_{t})||^2$ in Eq. 6. This is what we refer to in l. 124, when mentioning that we will pick $y_{t}$ such that $||\nabla f(x_{t})-\nabla f(y_{t})||^2$ (i.e. the discrepancy from SAM) can be controlled.
>
>
> > Q2.2 Does SAMPa-$\lambda$ also enjoy theoretical guarantees?
>
> A2.2. We have convergence gaurantees for $\lambda=0$, but not for more general $\lambda$. However, we believe that it should be possible to generalize the result considering that SAMPa-$\lambda$ is a convex combination of two convergent schemes as comment on in l. 136-138.
>
> We would like to emphasize that the empirical contribution is valuable in itself regarding SAMPa-0.2: the method _consistently_ outperforms SAM with 2$\times$ speedup across models and datasets (see Section 5). Note that this implies that it is possible to benefit from this method even without any parallelization. Specifically, one can run SAMPa-0.2 on a single GPU, in which case it would have the same running time as SAM, but importantly _still lead to superior performance_.
>
> > Q3. More on parallelization should be discussed. For example, how does the memory consumption of SAMPa, and how does it compare to other efficient SAM variants? How is the communication of SAMPa? Can communication be overlapped with computation? These questions should also be discussed in detail as the parallelization in SAMPa is the key novelty.
>
> A3. Thanks for the suggestion. We kindly remind that we have discussed the memory usage of SAMPa compared with SAM in Appendix E. We will give more detailed discussions about memory usage and communication overhead below.
> - **Memory usage:** In Appendix E, we mention that the memory usage of SAMPa on each GPU is the same with SAM's and note that SAMPa requires two GPUs while SAM needs one. For empirical results, we give memory usage comparison for all SAM variants in Section 5.2 below. SAMPa-0.2 even requires a little less memory usage on _each_ GPU, MESA requires roughly 23% extra memory compared with SAM, and other three methods have similar memory usage to SAM.
>
> |   | SAM | SAMPa-0.2 | LookSAM | AE-SAM |MESA| ESAM|
> |:---:|:---:|:---:|:---:|:---:|:---:|:---:|
> | Each GPU memory usage (MiB)| 2290 | 2016|2296| 2292 | 2814 | 2288 |
>
> - **Communication overhead:** In the experiment of Section 5.2, SAMPa's training time per epoch is 10.94s including 0.82s of communication overhead, compared to SAM's 18.81s. This indicates that SAMPa could approach a 2$\times$ speedup with optimized communication. Moreover, communication overhead might be overlapped by data loading, and further research into this could be interesting.
>
> > Q4. Can the authors comment more on the intuition on the reason that SAMPa outperforms SAM?
>
> A4. The method that consistently beats SAM is SAMPa-0.2, which is simply a convex combination of SAMPa and OptGD (see l. 138 for details). One hypothesis we currently have for the superior performance is that optimistic gradient descent might be better suited for nonsmooth parts of the optimization landscape, thereby stabilizing the algorithm when Lipschitz continuity assumption is not present (this assumption is crucial for allowing a fixed perturbation size). It is an interesting direction to understand this superior performance better.
>
>
> > Q5. Three typos.
>
> A5. Thanks for noticing the typo. We'll revise them in the final version.
>
> - In Algorithm 1, it should be $g_0=\nabla f(y_0, \mathcal{B}_0)$ at initialization.
> - In line 126, revise "identical" to "similar".
> - In line 180, revise to "we can trivially allow".

---

> > ### Comment · Reviewer_LruU · 2024-08-12
> >
> > Thanks for the detailed responses.
> >
> > I am still skeptical about A2.1, because i) the gradient norm might be too weak to measure the convergence in convex case; ii) convex problems are not the most suitable testbed for SAM.
> >
> > On the other hand, I do appreciate the additional experiments. This is reflected through the increased score.

---

> ### Author Response · Authors · 2024-08-13
> **Further response to Reviewer LruU**
>
> We sincerely thank the reviewer for their positive feedback and for raising the score.
>
> Regarding the two comments:
> - **Convergence of gradient norm:** We would like to emphasize that the goal is not to establish optimal rates for e.g. function value, but rather to match the rates of SAM, which we have demonstrated (please see the rebuttal for more elaboration). If convergence of function value is deemed important, note that it converges asymptotically due to the established descent like inequality.
> - **Convexity and SAM:** We do not understand why convex analysis should be less suitable for SAM than e.g. the mentioned AdaGrad or Adam. SAM has mostly seen success in overparameterized settings (where the training data is eventually perfectly fitted), where it has been exploited in several works that a convexification can occur. Additionally, the solution set for the convex problem is not necessarily a singleton, so biasing the obtained solution still seems reasonable.
>
> We thank again the reviewer for the positive engagement and remain available if there are any further questions.

---

### Official Review · Reviewer_c6Fm · 2024-07-08

**Soundness:** 3
**Presentation:** 2
**Contribution:** 2
**Rating:** 5
**Confidence:** 4

**Summary:**

This paper proposes a parallelized algorithm of Sharpness Aware Minimization(SAM) named SAMPa, which aims at making optimization more efficient by parallelizing the computation of one update in SAM. Plain SAM requires a 2x computational cost since 2 gradient computations (one for the perturbation and one for updating parameters) are needed at each training step. This paper utilizes an auxiliary sequence and another GPU device to calculate the perterbutation and gradient for updating parameters at the same time. Moreover, the convex analysis guarantees the convergence with a fixed perturbation size. SAMPa is shown to perform nearly as well as SAM on benchmark tasks such as, CIFAR10, CIFAR100, ImageNet-1k, and GLUE.

**Strengths:**

- The paper is well writen and mostly free of typos.
- The motivation of this paper is meaningful for practical application.
- Theoretical analysis supports the proposed method.
- The experiments are abundant and relatively comprehensive.

**Weaknesses:**

- This paper mentions the efficiency and effectiveness of SAM, the following paper ["Make Sharpness-Aware Minimization Stronger: A Sparsified Perturbation Approach"](https://proceedings.neurips.cc/paper_files/paper/2022/hash/c859b99b5d717c9035e79d43dfd69435-Abstract-Conference.html) should be considered to discuss in this paper. A test accuracy of comparison between SAMPa and this sparsified approach and a discussion in introduction would be helpful.
- A difference between the approximated perturbation based on auxiliary sequence $y_t$ and the real perturbation would be helpful.
- A Visualization of landscape and A spectrum of the Hessian(Figure 3 in ["SHARPNESS-AWARE MINIMIZATION FOR EFFICIENTLY
IMPROVING GENERALIZATION"](https://arxiv.org/pdf/2010.01412) and Figure 5 in ["Make Sharpness-Aware Minimization Stronger: A Sparsified Perturbation Approach"](https://proceedings.neurips.cc/paper_files/paper/2022/hash/c859b99b5d717c9035e79d43dfd69435-Abstract-Conference.html)) would be helpful for presenting the generalization superiority of SAMPa.
- The convex assumption of analysis is too strong, so I'm concerned that its results would have little significance for practical operation.
- The main reason why I hesitate to give a high score is that the paper lacks some intuitive or numerical explanation or analysis of "why SAMPa works" or "why we choose $y_t=...$ "

**Questions:**

See weakness.

**Limitations:**

The SAMPa requires 2 GPUs to realizing parallel computation, which will suffer communication cost and require more memory storage for auxiliary sequence.

---

> ### Author Rebuttal · Authors · 2024-08-07
>
> We thank the reviewer for their valuable feedback and address all remaining concerns below:
>
> > Q1. "Make Sharpness-Aware Minimization Stronger: A Sparsified Perturbation Approach" should be considered.
>
> A1. We have conducted additional experiments on Sparsed SAM (SSAM) below and will include them in our paper for completeness, but note that we have already compared against _four_ efficient variants of SAM in Section 5.2.
>
> SSAM perturbs a subset of weights to achieve computational gains. There are two methods in the SSAM paper: SSAM-F uses Fisher information for sparse perturbation, and SSAM-D uses efficient binary mask generation. We choose SSAM-D with 95% sparsity for our experiments due to the high computation cost of SSAM-F's empirical Fisher. Importantly, SAMPa-0.2 still shows a significant speedup compared to SSAM-D.
>
> ||SAM|SAMPa-0.2|SSAM-D|
> |:---:|:---:|:---:|:---:|
> |Time/Epoch (s)|18.81|10.94|16.95|
> |Accuracy (%)|94.26|94.62|94.48|
>
> Another sparse SAM variant, SAM-ON that only perturbs normalization layers, has been discussed in Sections 5.5 and 6. Note that both SSAM and SAM-ON can be incorporated within SAMPa for further improvement.
>
> > Q2. A difference between the approximated perturbation based on auxiliary sequence $y_t$ and the real perturbation would be helpful.
>
> A2. We claim that the similarity between the gradient on the auxiliary sequence $\nabla f(y_t)$ and the gradient on actual weight $\nabla f(x_t)$ holds not just in theory (Section 4) but also in practice.
>
> In the [figures](https://imgur.com/a/6hWQm44), we track the cosine similarity and Euclidean distance between $\nabla f(y_t)$ and $\nabla f(x_t)$ during whole training process of Resnet-56 on CIFAR-10. We find that the cosine similarity keeps above 0.99 during the whole training process, and in most period it's around 0.998, while at the end of training it is even close to 1. This indicates that SAMPa's estimated perturbation $\rho \frac{\nabla f(y_t)}{||\nabla f(y_t)||}$ is an excellent approximation of SAM's perturbation $\rho \frac{\nabla f(x_t)}{||\nabla f(x_t)||}$.
>
> Moreover, the Euclidean distance decreases and is close to zero at the end of training. This matches our theoretical analysis that $||\nabla f(x_t) -\nabla f(y_t)||^2$ eventually becomes small, which Lemma 4.3 guarantees in the convex case by establishing decrease of the potential function $\mathcal V_t$.
>
> > Q3. A spectrum of the Hessian would help.
>
> A3. Thanks for the suggestion. We provide [Hessian spectra](https://imgur.com/a/SFA1Ndf) for the Resnet-56 trained with SAM, SAMPa-0 and SAMPa-0.2 on CIFAR-10. We focus on the largest eigenvalue $\lambda_1$ and the ratio of the largest to the fifth largest eigenvalue $\frac{\lambda_1}{\lambda_5}$ as suggested by the reviewer, which reflect the flatness of the solution. As expected, the model trained with SAMPa-0 converges to minima with similar curvature to SAM.
>
> ||SAM|SAMPa-0|SAMPa-0.2|
> |:---:|:---:|:---:|:---:|
> |$\lambda_1$|94.29|87.17|149.63|
> |$\frac{\lambda_1}{\lambda_5}$|2.15|1.84|3.30|
>
> The model trained with SAMPa-0.2, on the other hand, converges to a sharper minimum while it shows better generalization performance. The existence of a well-performing sharper minimum is maybe not too surprising, considering that the relationship between sharpness and generalization remains unclear. For instance, [Andriushchenko and Flammarion, 2022] shows that a flatter minimum does not always lead to better generalization, which is further supported by [Mueller et al., 2024] and [1].
>
> > Q4. The convex assumption is too strong, so I'm concerned its results have little practical significance.
>
> A4. We kindly disagree that convex analysis is too strong for guiding practical applications. Arguably most of the successful optimizers today for deep learning were developed under the assumption of convexity. This includes AdaGrad, Adam and even more recent developments such as Distributed Shampoo and the Scheduler-free optimizer that won the AlgoPerf competition this month [2].
>
> Similarly, we found convex analysis extremely helpful in developing the algorithm. In our particular case our algorithmic design _falls out of the analysis_, specifically leading to our choice of the auxiliary sequence $(y_t)_{t\in \mathbb N}$ (see Section 3). We subsequently show that SAMPa indeed has strong practical performance in deep learning applications (see Section 5).
>
> > Q5. The main reason why I hesitate to give a high score is that the paper lacks some intuitive or numerical explanation or analysis of "why SAMPa works" or "why we choose $y_t=\dots$"
>
> A5. The particular choice of $y_t$ is a direct consequence of the analysis! Specifically, in Eq. 7 of the proof, the choice $y_{t+1} = x_t - \eta_t \nabla f(y_t)$ allows us to produce the term $||\nabla f(x_{t+1})-\nabla f(y_{t+1})||^2$ in order to telescope with $||\nabla f(x_{t})-\nabla f(y_{t})||^2$ in Eq. 6. This is what we refer to in l. 124, when mentioning that we will pick $y_{t}$ such that $||\nabla f(x_{t})-\nabla f(y_{t})||^2$ (i.e. the discrepancy from SAM) can be controlled. This gives a precise guarantee explaining why $\nabla f(x_{t})$ can be replaced by $\nabla f(y_{t})$.
>
> Empirically, the small difference between perturbations based on $\nabla f(x_t)$ and $\nabla f(y_t)$ (as discussed in Q2), along with similar Hessian spectra between SAM and SAMPa-0 (as shown in Q3) indicates that $\nabla f(y_t)$ is an excellent approximation of $\nabla f(x_t)$ in practice as well. Moreover, similar performance between SAMPa-0 and SAM supports our choice of $y_t$.
>
> We will highlight Eq. 7 in the final version and include a discussion on the additional empirical results above.
>
> [1] Ankit Vani, Frederick Tung, Gabriel L. Oliveira, Hossein Sharifi-Noghabi. Forget sharpness: perturbed forgetting of model biases within SAM dynamics. In International Conference on Machine Learning (ICML), 2024.
>
> [2] https://mlcommons.org/2024/08/mlc-algoperf-benchmark-competition/

---

> > ### Comment · Reviewer_c6Fm · 2024-08-14
> >
> > Thanks for the author's reply, I appreciate the detailed response. However, based on the concern about motivation and theory limitation, I would give my final score as Boardline Accept.

---

> ### Author Response · Authors · 2024-08-14
> **Further response to Reviewer c6Fm**
>
> We sincerely thank the reviewer for the response. We would like to address the remaining concerns:
>
> > 1. Motivation
>
> We kindly ask in what way our response for Q5 in the rebuttal did not clarify the motivation.
>
> In rebuttal, we explained that the particular choice of $y_{t+1} = x_t - \eta_t \nabla f(y_t)$ is a direct consequence of the analysis, which ensures that the discrepancy between $\nabla f(x_{t})$ and $\nabla f(y_{t})$ can be controlled. Empirically, the small difference between $\nabla f(x_t)$ and $\nabla f(y_t)$, along with similar Hessian spectra and test accuracy between SAM and SAMPa-0, confirms that $\nabla f(y_t)$ is an excellent approximation of $\nabla f(x_t)$.
>
> Additionally, Section 2.2 and Section 3 provide detailed explanations of our intuition and the development of SAMPa. Key points include:
> - *Perturbation direction matters:* RandSAM, which uses random perturbation, performs worse than SAM, highlighting the importance of perturbation direction.
> - *Using past gradients is ineffective:* OptSAM, using past gradients for perturbation, underperforms than SAM and it even fails to converge in a toy example due to its reliance on a poor gradient estimate from an ascent step.
> - *Introduce the auxiliary sequence:* We finally propose an auxiliary sequence $y_{t+1} = x_t - \eta_t \nabla f(y_t)$ as a direct result of our theoretical analysis, which aligns with the descent direction from $x_t$.
>
>
>
> > 2. Theory limitation
>
> We do not understand why convex analysis should be less suitable for SAM than e.g. the mentioned AdaGrad or Adam. SAM has mostly seen success in overparameterized settings (where the training data is eventually perfectly fitted), where it has been exploited in several works where convexification can occur. Additionally, the solution set for the convex problem is not necessarily a singleton, so biasing the obtained solution still seems reasonable.
>
>
> We hope our clarification addresses all your concerns, and we are happy to discuss them if you have any further questions.
>
> We would like to emphasize that SAMPa significantly advances SAM by breaking the sequential nature of its two gradient computations, enabling parallel execution. SAMPa not only maintains convergence guarantees in the convex setting but also, as shown in our empirical results, achieves the fastest computational time among five efficient SAM variants (including SSAM introduced in the rebuttal) while consistently improving generalization across multiple tasks. We kindly ask the reviewer to consider these contributions.
>
> Thank you again for your engagement!

---

### Official Review · Reviewer_zVXP · 2024-07-13

**Soundness:** 3
**Presentation:** 3
**Contribution:** 3
**Rating:** 7
**Confidence:** 3

**Summary:**

This paper proposes a modification of SAM, named SAMPa, which enables to fully parallelize the two gradient computations in SAM, in order to accelerating the training. By doubling the computation resources, parallelized SAM could approach to a twofold speedup of SAM. Theoretical analysis is provided for the convergence of the proposed algorithm. Empirical results show that SAMPa can match or even surpass SAM in test accuracy.

**Strengths:**

1. This paper proposes a modification of SAM, named SAMPa, which enables to fully parallelize the two gradient computations in SAM, in order to accelerating the training. By doubling the computation resources, parallelized SAM could approach to a twofold speedup of SAM.

2. Theoretical analysis is provided for the convergence of the proposed algorithm.

3. Empirical results show that SAMPa can match or even surpass SAM in test accuracy.

**Weaknesses:**

1. To approach to a twofold speedup of SAM, SAMPa requires double of the computation resources compare to SAM.

2. There are previous works actually reduce the overall computation overhead of SAM. I think these works should also be compared to SAMPa as baselines in the experiments. For example:
Du, J., Zhou, D., Feng, J., Tan, V.Y., & Zhou, J.T. (2022). Sharpness-Aware Training for Free. NeurIPS 2022.

--------
Most of my concerns are addressed according to the author's feedback.

**Questions:**

1. Is there any theoretical explanation about why SAMPa could outperform SAM in test accuracy?

**Limitations:**

The limitations are well discussed in this paper.

---

> ### Author Rebuttal · Authors · 2024-08-07
>
> We thank the reviewer for their valuable feedback and address all remaining concerns below:
>
> > Q1. To approach to a twofold speedup of SAM, SAMPa requires double of the computation resources compare to SAM.
>
> A1. Please note that the total computational time for SAMPa across all GPUs is comparable to that of SAM. In addition, we would like to emphasize that SAMPa-$\lambda$ _consistently_ outperforms SAM across models and datasets (see Section 5) with 2$\times$ speedup. It can show advantages even without parallelization. Specifically, one run SAMPa-$\lambda$ on a single device has the same running time as SAM, but importantly _still leads to superior performance_.
>
> > Q2. There are previous works actually reduce the overall computation overhead of SAM. I think these works should also be compared to SAMPa as baselines in the experiments. For example: Du, J., Zhou, D., Feng, J., Tan, V.Y., & Zhou, J.T. (2022). Sharpness-Aware Training for Free. NeurIPS 2022.
>
> A2. We kindly remind that in Section 5.2, we have already considered one of two methods named MESA in "Sharpness-Aware Training for Free" as a baseline. For completeness, we also add result for SAF in the table below. Although SAF requires slightly less training time, its performance is worse than the other three methods. SAMPa-0.2 can reach the same accuracy as SAF only with 75% of SAF's epochs. Furthermore, SAMPa's time per epoch is 10.94s including 0.82s of communication overhead, which can be reduced further with faster device communication.
>
> |   | SAM |SAMPa-0.2 | MESA | SAF |
> |:---:|:---:|:---:|:---:|:---:|
> | Time/Epoch (s) |  18.81  |  10.94 | 15.43 | 10.09 |
> | Accuracy (%) | 94.26 | 94.62 | 94.23  | 93.89 |
>
> > Q3. Is there any theoretical explanation about why SAMPa could outperform SAM in test accuracy?
>
> A3. The method that consistently beats SAM is SAMPa-0.2, which is simply a convex combination of SAMPa and OptGD (see l. 138 for details). One hypothesis we currently have for its superior performance is that optimistic gradient descent might be better suited for nonsmooth parts of the optimization landscape, thereby stabilizing the algorithm when Lipschitz continuity assumption is not present (this assumption is crucial for allowing a fixed perturbation size). It is an interesting direction for future work to understand this superior performance better.

---

> > ### Comment · Reviewer_zVXP · 2024-08-12
> >
> > I thank the authors for their feedback. Most of my concerns are addressed and I would like to raise the score.
> >
> > After reading the author's feedback, it brings to my attention that SAMPa-0.2 is the overall best which is basically a convex combination of ordinary gradients and SAM gradients.
> > Note that there is actually a variant of SAM that does sth. similar:
> > Zhao, Y., Zhang, H., & Hu, X. Penalizing gradient norm for efficiently improving generalization in deep learning. ICML 2022.
> >
> > I think the gradient penalization method above with tuned interpolation ratio would be a better baseline than SAM (and I'm sorry for bringing this up so late).
> > However, since SAMPa-0 also beats SAM in some cases, it seems that the improvement of SAMPa comes not only from the interpolation.

---

> > > ### Author Response · Authors · 2024-08-13
> > > **Further response to Reviewer zVXP**
> > >
> > > We sincerely thank the reviewer for the positive response and for increasing the score.
> > >
> > > Thank you for bringing the paper about penalizing gradient norms to our attention. The convex combination of gradients on ordinary and perturbed weights in that method is indeed similar to SAMPa-$\lambda$, and we agree this may contribute to the performance improvement.
> > >
> > > However, it's important to note that SAMPa-$\lambda$ differs in a key aspect: it computes gradients for each update on _two different batches_ (as shown in line 6 of Algorithm 1), while the penalizing method combines gradients from the same batch.
> > >
> > > We conducted preliminary experiments using the penalizing method on CIFAR-10 with the same hyperparameters as SAMPa-0.2. The results suggest similar performance in standard classification tasks, but worse outcomes with noisy labels. We would like to explore this further in future work.
> > >
> > > |   | SAM |SAMPa-0.2 | Penalizing |
> > > |:---:|:---:|:---:|:---:|
> > > | Resnet-56 |  94.26  |  94.62 | 94.57 |
> > > | Resnet-32 (80% noisy label) | 48.01 | 49.92 | 48.26 |
> > >
> > >
> > > As mentioned in the rebuttal, another potential reason for SAMPa’s enhanced performance is that optimistic gradient descent may be better suited for nonsmooth optimization landscapes.
> > >
> > > We will include this discussion in the final version. Thank you again for your positive engagement, and we remain available for any further questions.

---

### Author Response · Authors · 2024-08-12
**Happy to provide further clarification**

Dear reviewers,

Thank you for your constructive feedback. During the rebuttal, we addressed the following key points, with detailed responses in the rebuttal:

- **Choice of $y_t$:** The choice of $y_{t+1} = x_t - \eta_t \nabla f(y_t)$ enables telescoping in our analysis. Empirically, $\nabla f(y_t)$ closely approximates $\nabla f(x_t)$, supported by small differences and similar Hessian spectra between SAM and SAMPa-0.
- **Why SAMPa-0.2 outperforms SAM:** We hypothesize that the inclusion of optimistic gradient descent in SAMPa-0.2 stabilizes the algorithm in nonsmooth optimization landscapes, improving performance when Lipschitz continuity is not assumed.
- **Convex analysis:** We emphasize that convex analysis has been foundational for many successful deep learning optimizers and was instrumental in our design of the auxiliary sequence $y_t$.
- **Efficient SAM variants:** Additional experiments comparing SAMPa with _SAM for Free_ and _Sparsified SAM_ further demonstrate its improvements in both generalization and efficiency.
- **Extra resource:** SAMPa-$\lambda$ consistently outperforms SAM across models and datasets, achieving a 2$\times$ speedup with parallelization, and still delivers superior performance even on a single device.
- **Data parallelization:** Our results show that data parallelization slows down training, underscoring the importance of gradient parallelization.
- **Memory and communication:** SAMPa's memory usage per GPU is comparable to SAM's. Its communication overhead could potentially be reduced or hidden.


As the discussion window is closing soon, if you have any additional questions, we would be happy to elaborate on them. Otherwise, we kindly request you to consider increasing the score. Thank you!

Best regards,

Authors

---

### Decision · Program_Chairs · 2024-09-25

**Decision:**

Accept (poster)

**Comment:**

This paper presents SAMPa, a parallelized variant of Sharpness-Aware Minimization (SAM) aimed at improving the efficiency and generalization of deep learning models. The key innovation is the introduction of an auxiliary sequence to parallelize the two gradient computations in SAM, which traditionally incurs double the per-iteration computational cost compared to standard optimizers like SGD.
The empirical results are strong, and the method shows significant potential for improving the efficiency of SAM. Though this paper does not provide any theoretical explanation on why SAMPa can have better generalization, SAMPa is a practically valuable contribution to the field of deep learning optimization. The reviewers still unanimously recommends accepting this paper. The authors should include the additional experiments in the rebuttal into the final version, especially those related to mSAM and recent other baseline methods.

Minor comments:
The main theorem (Lemma 4.3 and Theorem 4.4) has a strange coefficient $(1+L+L^2)$. It is strange because $L$ is not unit-free and should not be added directly across different powers. By a quick glance it seems to be caused by typos in proofs, e.g., the third line in equation (6), $(1-\eta_t)$ should be $\eta_t$.